# Zero-cost Proxy for Adversarial Robustness Evaluation

**Yuqi Feng, Yuwei Ou, Jiahao Fan, Yanan Sun**[*]
College of Computer Science, Sichuan University
`feng770623@gmail.com; ouyuwei@stu.scu.edu.cn;`
`{fanjh,ysun}@scu.edu.cn`

## Abstract

Deep neural networks (DNNs) easily cause security issues due to the lack of adversarial robustness. An emerging research topic for this problem is to design adversarially robust architectures via neural architecture search (NAS), i.e., robust NAS. However, robust NAS needs to train numerous DNNs for robustness estimation, making the search process prohibitively expensive. In this paper, we propose a zero-cost proxy to evaluate the adversarial robustness without training. Specifically, the proposed zero-cost proxy formulates the upper bound of adversarial loss, which can directly reflect the adversarial robustness. The formulation involves only the initialized weights of DNNs, thus the training process is no longer needed. Moreover, we theoretically justify the validity of the proposed proxy based on the theory of neural tangent kernel and input loss landscape. Experimental results show that the proposed zero-cost proxy can bring more than $20\times$ speedup compared with the state-of-the-art robust NAS methods, while the searched architecture has superior robustness and transferability under white-box and black-box attacks. Furthermore, compared with the state-of-the-art zero-cost proxies, the calculation of the proposed method has the strongest correlation with adversarial robustness. Our source code is available at `https://github.com/fyqsama/Robust_ZCP`.

## 1 Introduction

Deep neural networks (DNNs) are vulnerable to adversarial attacks (Szegedy et al., 2014), bringing potential hazards to security-critical systems that apply DNNs (Eykholt et al., 2018; Sharif et al., 2016). To address this problem, an emerging topic is to design adversarially robust DNNs via neural architecture search (robust NAS) (Ou et al., 2024). Current robust NAS methods need to evaluate the adversarial robustness of a large number of DNNs to select out the robust ones. However, the mainstream evaluation methods (e.g., training from scratch or weight-sharing technique (Pham et al., 2018)) are inefficient due to the training process of DNNs, and this problem becomes sharper when performing adversarial training to evaluate the adversarial robustness (Goodfellow et al., 2014). This is because the adversarial examples are generated during adversarial training, making it much more time-consuming than the standard training which does not need to generate adversarial examples.

Recently, a few works (Lukasik et al., 2023; Ha et al., 2024) try to reduce the evaluation cost via zero-cost proxies (Abdelfattah et al., 2021). Specifically, zero-cost proxies are surrogate tasks evaluating DNNs without training. Zero-cost proxies were initially designed for natural accuracy evaluation, but now a few works also try to use them for evaluating adversarial robustness. For example, Lukasik *et al.* (Lukasik et al., 2023) show that some zero-cost proxies designed for natural accuracy evaluation can also be used for adversarial robustness evaluation. Ha *et al.* (Ha et al., 2024) design another zero-cost proxy, i.e., CRoZe, to evaluate the adversarial robustness. Although these works have taken the first step towards zero-cost proxies for adversarial robustness evaluation, they still require adversarial example generation and lack theoretical analysis, increasing the computational cost and making the users do not know why the proxies can reveal robustness. In order to further accelerate the adversarial robustness evaluation with theoretical guarantee, we design a robust zero-cost proxy without adversarial example generation along with theoretical justifications in this paper.

---

[*]Corresponding author.

Specifically, we propose a zero-cost proxy that describes the upper bound of adversarial loss (i.e., loss on adversarial examples), which is negatively correlated to adversarial robustness. Meanwhile, the upper bound of adversarial loss is represented by the initialized weights of DNNs, and can be directly used to evaluate adversarial robustness without adversarial example generation. Furthermore, we mathematically prove the validity of the proposed zero-cost proxy based on the theory of neural tangent kernel (NTK) (Jacot et al., 2018) and input loss landscape (Zhao et al., 2020).

In order to show the efficiency and effectiveness of the proposed zero-cost proxy, we perform the proposed zero-cost proxy along with NAS, searching for adversarially robust architectures. Surprisingly, when the proposed zero-cost proxy is adopted, there is a $20\times$ speedup compared with the state-of-the-art robust NAS methods (Dong et al., 2020; Hosseini et al., 2021; Cheng et al., 2023), while the derived architectures are still competitive under both white-box and black-box attacks.

Moreover, we comprehensively analyze the correlation between the output of the proposed zero-cost proxy and the adversarial robustness on a recently constructed dataset (Jung et al., 2023), which we call NAS-Bench-201-R for convenience. NAS-Bench-201-R is the first to evaluate a full NAS-Bench-201 search space (Dong & Yang, 2020) under various adversarial attacks with various parameters, and it can be well used for evaluating the performance of the zero-cost proxy. However, NAS-Bench-201-R can only be used to validate the zero-cost proxy when attacks are weak, it does not work when attacks are strong. This is because their networks are not adversarially trained, and their adversarial accuracy may drop to near zero when suffering strong attacks. Such data is not informative. Therefore, to evaluate the zero-cost proxy more comprehensively, we construct a dataset called Tiny-RobustBench, mapping adversarially trained networks to their adversarial accuracy. Experimental results on NAS-Bench-201-R show that the proposed zero-cost proxy performs well when the adversarial accuracy is practically meaningful (i.e., not drop to near zero). Furthermore, experimental results on Tiny-RobustBench show that the output of the proposed zero-cost proxy has the strongest correlation to adversarial robustness among state-of-the-art zero-cost proxies.

Our contributions can be summarized as follows: 1) We propose a zero-cost proxy to evaluate the adversarial robustness without training, and the relationship between the proxy and adversarial robustness is theoretically proven. 2) We apply the proposed zero-cost proxy to NAS, and the search cost is dozens of times less than state-of-the-art robust NAS methods, while the searched architecture is competitive in terms of white-box attacks, black-box attacks, and transferability. Massive outstanding architectures are analyzed to guide the future design of adversarially robust architectures. 3) We comprehensively evaluate the correlation between the output of the proposed zero-cost proxy and the adversarial robustness under different adversarial attacks in different search spaces. Moreover, a novel dataset named Tiny-RobustBench is open sourced to promote the development of this field.

## 2 RELATED WORK

### 2.1 ADVERSARIAL ATTACKS AND DEFENSES

According to whether or not attackers have full access to DNN parameters, adversarial attacks can be divided into white-box and black-box attacks. Common white-box attacks include FGSM (Goodfellow et al., 2014), PGD (Madry et al., 2018), BIM (Kurakin et al., 2018), C&W (Carlini & Wagner, 2017), and so on. One commonly used black-box attack is the transfer-based attack (Papernot et al., 2017), which generates adversarial examples utilizing a substitute DNN. Recently, AutoAttack (Croce & Hein, 2020), which is an ensemble of white-box and black-box attacks, becomes popular for a fair adversarial robustness evaluation.

To defend DNNs from adversarial attacks, the adversarial defense techniques have emerged. One popular defense technique is adversarial training (Goodfellow et al., 2014). Other adversarial defense techniques include defensive distillation (Papernot et al., 2016), obfuscated gradients (Athalye et al., 2018), data compression (Dziugaite et al., 2016), and so on. In this paper, we apply the adversarial training for its effectiveness, and test the adversarial accuracy under the adversarial attacks.

### 2.2 NEURAL TANGENT KERNEL (NTK) AND ROBUSTNESS

NTK (Jacot et al., 2018) is defined as the Gram matrix of gradients, and can be used to analyze the convergence and generalization of the model. There have been some studies investigating the

model robustness based on the NTK theory. For example, Wu *et al.* (Wu et al., 2021) study the relationship between network width and adversarial robustness, and explain it based on NTK. Tsilivis *et al.* (Tsilivis & Kempe, 2022) study how to generate adversarial examples using NTK. Singla *et al.* (Singla et al., 2021) show that shift invariance may reduce adversarial robustness based on the NTK theory. Gao *et al.* (Gao et al., 2019) show some properties of adversarial training, which are also proven based on NTK. However, these studies are still far from our work. We aim to conduct quantitative analysis to compare the adversarial robustness of different architectures, especially when different primitives (e.g., convolutional and pooling layers) and topologies (how to connect different layers) are involved, while previous studies could not.

## 2.3 ZERO-COST PROXY

Zero-cost proxy is a technique that evaluates DNNs using a substitute task requiring little computational cost. For instance, Abdelfattah *et al.* (Abdelfattah et al., 2021) study a series of zero-cost proxies including grad_norm, snip, grasp, synflow, fisher, and jacob_cov (Mellor et al., 2021). Zhang *et al.* (Zhang & Jia, 2021) propose GradSign, which predicts the DNN performance through local optima density. Recently, some advanced zero-cost proxies have attracted much attention for their strong interpretability. Their correlations to the final performance of DNNs are mathematically proven, based on the theory of NTK. For example, Xu *et al.* (Xu et al., 2021) prove that the mean of the NTK matrix is negatively correlated to the training loss, and propose MGM as a zero-cost proxy. Shu *et al.* (Shu et al., 2022a) also derive the upper bound of training loss based on NTK and use it as a zero-cost proxy. Besides, Shu *et al.* (Shu et al., 2022b) develop a novel framework that consistently boosts existing zero-cost proxies.

The aforementioned zero-cost proxies are designed for the prediction of natural accuracy. Recently, Lukasik *et al.* (Lukasik et al., 2023) show that these zero-cost proxies can also be used for adversarial robustness evaluation. Ha *et al.* (Ha et al., 2024) design another zero-cost proxy, named CRoZe, to evaluate the robustness of architectures against diverse perturbations. Although these works have taken the first step towards zero-cost proxies for adversarial robustness evaluation, they still need to costly generate adversarial examples and do not provide theoretical analysis about why their proxies can reveal robustness. In this paper, we aim to design a zero-cost proxy which is theoretically guaranteed and no longer needs adversarial example generation, thus the efficiency is further achieved.

## 3 METHODOLOGY

### 3.1 PRELIMINARIES

We present the preliminaries of NTK and input loss landscape, which form the basis of the proposed zero-cost proxy.

**Proposition 1** (Xu et al. (2021)). *Suppose $f_{\theta_t} : \mathbb{R}^d \to \mathbb{R}$ defines a DNN, with its weight parameters $\theta$ at the t-th epoch. Suppose $x$ and $y$ denote image samples and the corresponding label vector, respectively. For any $t > 0$, Eq. (1) holds:*

$$\|y - f_{\theta_t}(x)\|_2^2 \leq \exp(-\lambda_{min}(\Theta_{\theta_t})t)\|y - f_{\theta_0}(x)\|_2^2, \tag{1}$$

*where $\lambda_{min}(\Theta_{\theta_t})$ is the minimum eigenvalue of the NTK $\Theta_{\theta_t}$. The NTK is essentially the dot product of two gradient vectors with different input samples $x$ and $x'$, formulated as Eq. (2):*

$$\Theta_{\theta_t}(x, x') = \langle \nabla_\theta f_{\theta_t}(x), \nabla_\theta f_{\theta_t}(x')^\top \rangle, \tag{2}$$

*where $\nabla_\theta f_{\theta_t}$ means to the Jacobian of DNN prediction.*

Mok *et al.* (Mok et al., 2022) show that Eq. (1) can be further transformed to Eq. (3) because the NTK remains constant regardless of training under some assumptions:

$$\|y - f_{\theta_t}(x)\|_2^2 \leq \exp(-\lambda_{min}(\Theta_{\theta_0})t)\|y - f_{\theta_0}(x)\|_2^2. \tag{3}$$

The assumptions involved in this transformation are detailed in Appendix A.1.

**Proposition 2** (Zhao et al. (2020)). *Suppose $f_\theta$ defines a DNN, with its weight parameters $\theta$. Suppose $\mathcal{L}(\theta, x)$ denotes the classification loss with weight parameters $\theta$ and input $x$. Consider the oracle*

*adversarial loss $max_{\|\delta\|\leq\epsilon}\mathcal{L}(\theta(t), x+\delta)$ of the model $t$ on the path connecting two independently trained models, where $\delta$ denotes a perturbation to $x$ confined by an $\epsilon$-ball induced by a vector norm $\|\cdot\|$. Let $c$ denote the normalized inner product in absolute value for the largest eigenvector $v$ of the input Hessian and $\nabla_x\mathcal{L}(\theta(t), x)$, i.e., $\frac{|\nabla_x\mathcal{L}(\theta(t),x)^\top v|}{\|\nabla_x\mathcal{L}(\theta(t),x)\|} = c$. Then under some assumptions, we have $max_{\|\delta\|\leq\epsilon}\mathcal{L}(\theta(t), x+\delta) \sim \lambda_{max}(H_t(x))$ as $c \to 1$.*

## 3.2 DESIGN OF THE ZERO-COST PROXY

In this section, we present the proposed zero-cost proxy for adversarial robustness evaluation, and the theoretical analysis is provided in the next section. The design of a zero-cost proxy for adversarial robustness evaluation would be more challenging compared with designing a zero-cost proxy for the natural accuracy evaluation, because the adversarial robustness is dependent on the natural accuracy and requires extra adversarial robustness-related calculation.

The proposed zero-cost proxy is formulated as Eq. (4):

$$R = -\exp\left(\frac{1}{MN^2}\sum_{m=1}^{M}\sum_{i=1}^{N}\sum_{j=1}^{N}\left(\frac{\partial f_{\theta_0}(x_i)}{\partial\theta_0^m}\right)\left(\frac{\partial f_{\theta_0}(x_j)}{\partial\theta_0^m}\right)^\top t\right) \times \left\|\frac{l(x+hz^*)-l(x)}{h}\right\|_2, \quad (4)$$

where $R$ is the calculation of the zero-cost proxy, $M$ is the number of layers, $N$ is the number of data samples, $\theta^m$ is the sampled parameters from the $m$-th layer, $l(x) = \nabla_x\mathcal{L}(\theta_0, x)$, $z^* = \frac{sign(\nabla_x\mathcal{L}(\theta_0,x))}{\|sign(\nabla_x\mathcal{L}(\theta_0,x))\|}$, and $h$ controls the scale of the loss landscape on which the smoothness is induced.

Eq. (4) can be directly used to efficiently evaluate the adversarial robustness of DNNs. The efficiency mainly comes from two aspects. First, the proposed zero-cost proxy only needs to iterate samples instead of generating adversarial examples. This is because the NTK is calculated only based on different samples $x$ and $x'$ as shown in Eq. (2). These samples can be obtained by simply iterating samples in a batch, thus there is no need to generate adversarial examples. Consequently, the proposed zero-cost proxy is computational efficient, because the repetitive calculation for the gradient or the time-consuming optimization for the adversarial example generation are no longer needed. Second, the proposed zero-cost proxy only needs the initial weights $\theta_0$ of the DNN. Because there is no training process included, the efficiency is further guaranteed.

## 3.3 THEORETICAL ANALYSIS

In this section, we carry out the theoretical analysis to justify the validity of the proposed zero-cost proxy. Specifically, our motivation comes from the upper bound of adversarial loss. This is because the upper bound of adversarial loss is negatively correlated to adversarial robustness, and can be used as a proxy for adversarial robustness evaluation. From this perspective, we consider the upper bound of adversarial loss $\|y - f_{\theta_t}(\hat{x})\|_2^2$, where $\hat{x}$ denotes the adversarial examples generated using $x$. By replacing $x$ in Eq. (3) with $\hat{x}$, Eq. (5) holds:

$$\|y - f_{\theta_t}(\hat{x})\|_2^2 \leq \exp(-\lambda_{min}(\hat{\Theta}_{\theta_0})t)\|y - f_{\theta_0}(\hat{x})\|_2^2, \quad (5)$$

where $\hat{\Theta}_{\theta_0}$ denotes the NTK matrix computed by adversarial examples $\hat{x}$ and $\hat{x}'$ generated using $x$ and $x'$. Similar to Eq. (2), the NTK matrix computed by adversarial examples can be formulated as Eq. (6):

$$\hat{\Theta}_{\theta_0}(\hat{x}, \hat{x}') = \langle\nabla_\theta f_{\theta_0}(\hat{x}), \nabla_\theta f_{\theta_0}(\hat{x}')^\top\rangle. \quad (6)$$

Eq. (5) shows that the final adversarial loss is upper bounded by the right term, which is negatively correlated to adversarial robustness and can be used for adversarial robustness evaluation. However, both the minimum eigenvalue of the NTK matrix $\lambda_{min}(\hat{\Theta}_{\theta_0})$ and the adversarial loss at initialization $\|y - f_{\theta_0}(\hat{x})\|_2^2$ in the right term require the generation of adversarial examples $\hat{x}$ for further calculation, which is computationally expensive and violates the principle of "zero-cost". To avoid the generation of adversarial examples, we are committed to finding two substitutes of $\lambda_{min}(\hat{\Theta}_{\theta_0})$ and $\|y - f_{\theta_0}(\hat{x})\|_2^2$.

For the substitute of the adversarial loss at initialization $\|y - f_{\theta_0}(\hat{x})\|_2^2$, Proposition 2 gives a feasible solution, i.e., $\|y - f_{\theta_0}(\hat{x})\|_2^2 \sim \lambda_{max}(H_{\theta_0}(x))$. The validity of this solution is discussed in detail in Section 3.5. For higher efficiency, following a previous research (Mok et al., 2021), we approximate

$\lambda_{max}(H_{\theta_0}(x))$ as Eq. (7):

$$\lambda_{max}(H_{\theta_0}(x)) \approx \left\| \frac{l(x + hz^*) - l(x)}{h} \right\|_2, \tag{7}$$

where $l(x) = \nabla_x \mathcal{L}(\theta_0, x)$, $z^* = \frac{\text{sign}(\nabla_x \mathcal{L}(\theta_0, x))}{\|\text{sign}(\nabla_x \mathcal{L}(\theta_0, x))\|}$, and $h$ controls the scale of the loss landscape on which the smoothness is induced.

As for the substitute of the minimum eigenvalue of the NTK matrix $\lambda_{min}(\hat{\Theta}_{\theta_0})$, we find that $\lambda_{min}(\Theta_{\theta_0})$ is positively correlated to natural accuracy but negatively correlated to adversarial accuracy. This could be explained by the competing relationship between both indicators (Tsipras et al., 2018; Zhang et al., 2019). More discussion and theoretical analysis about this phenomenon can be found in Section 4.2.3 and Appendix A.2. Therefore, we simply use $\lambda_{min}(\Theta_{\theta_0})$ to replace $-\lambda_{min}(\hat{\Theta}_{\theta_0})$, which avoids the generation of adversarial examples. Meanwhile, following a previous research (Xu et al., 2021), we calculate $\lambda_{min}(\Theta_{\theta_0})$ via Eq. (8) for a higher efficiency:

$$\lambda_{min}(\Theta_{\theta_0}) = \frac{1}{MN^2} \sum_{m=1}^{M} \sum_{i=1}^{N} \sum_{j=1}^{N} (\frac{\partial f_{\theta_0}(x_i)}{\partial \theta_0^m})(\frac{\partial f_{\theta_0}(x_j)}{\partial \theta_0^m})^\top, \tag{8}$$

where $M$ is the number of network layers, and $N$ is the number of data samples. $\theta^m$ is the sampled parameters from the $m$-th layer where the length of $\theta^m$ is 50 in implementation.

Finally, we replace $\|y - f_{\theta_0}(\hat{x})\|_2^2$ and $-\lambda_{min}(\hat{\Theta}_{\theta_0})$ in Eq. (5) with $\lambda_{max}(H_{\theta_0}(x))$ and $\lambda_{min}(\Theta_{\theta_0})$ formulated by Eq. (7) and Eq. (8), respectively. Meanwhile, we add a minus sign before the right term of Eq. (5) to make it positively correlated to adversarial robustness, which is consistent with most zero-cost proxies. Consequently, we obtain Eq. (4) provided in the previous subsection, which can be used to evaluate the adversarial robustness of DNNs.

### 3.4 COMPUTATIONAL COMPLEXITY ANALYSIS

The computational complexity of the proposed zero-cost proxy is determined by the number of both layers and data samples. Given the architecture with $M$ layers, because our proxy needs to traverse all layers once in the evaluation, the computational complexity for this step is $O(M)$. Meanwhile, because our proxy requires two traversals of the data for the calculation of each layer. Considering the number of data samples is $N$, the computational complexity for this step is $O(N^2)$. According to the above analysis, the overall computational complexity of the proposed zero-cost proxy is $O(MN^2)$.

### 3.5 DISCUSSION

In this section, we will justify the validation of adopting the largest eigenvalue $\lambda_{max}(H_{\theta_0}(x))$ to approximate the upper bound of adversarial loss. As empirically stated by Moosavi *et al.* (Moosavi-Dezfooli et al., 2019), $c$ in Proposition 2 is equal to 0.43 before adversarial fine-tuning, and it increases to 0.90 after the adversarial fine-tuning on CIFAR-10. It is demonstrated that the condition where $c$ is approaching 1.0 can be approximately satisfied in practice, especially for the architectures adversarially trained. As a result, adopting the largest eigenvalue $\lambda_{max}(H_{\theta_0}(x))$ to approximate the upper bound of adversarial loss is practical.

## 4 EXPERIMENTS

We study the following research questions:

**RQ1:** Is the proposed zero-cost proxy superior in robust accuracy and cost-efficiency?

**RQ2:** Is the proposed zero-cost proxy superior in evaluating adversarial robustness of architectures?

### 4.1 RQ1: SUPERIORITY IN TERMS OF ROBUST ACCURACY AND COST-EFFICIENCY

#### 4.1.1 EXPERIMENTAL SETTINGS

**Benchmark Datasets** Following the conventions of the robust NAS community (Guo et al., 2020; Mok et al., 2021; Ou et al., 2024), CIFAR-10 (Krizhevsky & Hinton, 2009), SVHN (Netzer et al.,

2011), Tiny-ImageNet-200 (Le & Yang, 2015), and ImageNet (Deng et al., 2009) are used as benchmark datasets.

**Peer Competitors**   We choose architectures commonly used in the robust NAS community (Mok et al., 2021; Ou et al., 2024) as our peer competitors, including ResNet-18 (He et al., 2016), DenseNet-121 (Huang et al., 2017), DARTS (Liu et al., 2019), PDARTS (Chen et al., 2019), RobNet-free (Guo et al., 2020), RACL (Dong et al., 2020), DSRNA (Hosseini et al., 2021), and WsrNet (Cheng et al., 2023). Besides, we also choose some training-free NAS methods, including GradNorm (Chen et al., 2018), SynFlow (Tanaka et al., 2020), and CRoZe (Ha et al., 2024). The experimental results of these training-free NAS methods are cited from reference (Ha et al., 2024).

**Parameter Settings**   Neural architectures are stacked by 20 cells, with an initial channel number of 36. After the search, the best architecture is adversarially trained. Following advanced settings in the robust NAS community (Mok et al., 2021; Ou et al., 2024), the adversarial training is performed using a seven-step PGD with a step size of 0.01 and a total perturbation of 8/255. SGD is used to optimize networks, with the momentum of 0.9 and the weight decay of $1 \times 10^{-4}$. The learning rate is set to 0.1 initially, and decayed by a factor of 0.1 at the 100-th epoch. The batch size is set to 64. The search cost is measured by GPU days (number of GPUs used $\times$ total running time (days)) using the NVIDIA RTX 2080Ti GPU.

### 4.1.2   PERFORMANCE UNDER WHITE-BOX ATTACKS

To apply the proposed zero-cost proxy to NAS and explore its performance under white-box attacks, we construct a simple NAS algorithm, randomly sampling 1,000 architectures from the DARTS search space, and adopting the architecture with the highest calculated score as our search result. The searched architecture is adversarially trained and evaluated under several white-box attacks. The results are shown in Table 1. The best and second-best results are in bold and underlined, respectively.

Table 1: Evaluation results of adversarially trained models on CIFAR-10 under white-box attacks.

| Category | Model | With Training? | Params (M) | FLOPs (M) | Natural Acc. (%) | FGSM (%) | PGD$^{7\dagger}$ (%) | PGD$^{20}$ (%) | PGD$^{100}$ (%) | APGD$_{CE}$ (%) | AA (%) | Search Cost (GPU Days) |
|---|---|---|---|---|---|---|---|---|---|---|---|---|
| Hand-Crafted | ResNet-18 | × | 11.2 | 37.67 | 84.09% | 54.64% | - | 45.86% | 45.53% | 44.54% | 43.22% | - |
| | DenseNet-121 | × | 7.0 | 59.83 | **85.95%** | 58.46% | - | 50.49% | 49.92% | 49.11% | 47.46% | - |
| Standard NAS | DARTS | ✓ | 3.3 | 547.44 | 85.17% | 58.74% | - | 50.45% | 49.28% | 48.32% | 46.79% | 1.0 |
| | PDARTS | ✓ | 3.4 | 550.75 | 85.37% | 59.12% | - | 51.32% | 50.91% | 49.96% | 48.52% | 0.3 |
| | GradNorm$^\dagger$ | × | 4.7 | - | 81.61% | - | 49.86% | - | - | - | 46.69% | 0.1 |
| | SynFlow$^\dagger$ | × | 5.1 | - | 77.08% | - | 45.95% | - | - | - | 42.45% | 0.1 |
| Robust NAS | RobNet-free | ✓ | 5.6 | 800.40 | 85.00% | 59.22% | - | 52.09% | 51.14% | 50.41% | 48.56% | >3.3 $^*$ |
| | RACL | ✓ | 3.6 | 568.86 | 83.97% | 59.29% | - | 52.18% | 51.72% | 51.24% | 48.59% | 0.5 |
| | DSRNA | ✓ | 2.0 | 336.23 | 80.93% | 54.49% | - | 49.11% | 48.89% | 48.54% | 44.87% | 0.4 |
| | WsrNet | ✓ | 3.0 | 484.30 | 83.94% | 56.12% | - | 47.17% | 46.61% | - | 43.91% | 4.0 |
| | CRoZe$^\dagger$ | × | 5.5 | - | 84.28% | - | 52.17% | - | - | - | 48.14% | 0.2 |
| | **Ours** | × | 3.4 | 555.54 | 85.60% | **60.20%** | **69.21%** | **52.75%** | **52.51%** | **52.25%** | **49.97%** | **0.017** |

$^*$ RobNet neither reported their search cost nor provided their search code, so we have to estimate their search cost according to their principles. Specifically, RobNet is based on the one-shot NAS method and requires extra search cost when generating adversarial examples for evaluation, so their search cost is estimated to be longer than 3.3 GPU Days of the one-shot NAS method (Bender et al., 2018).

$^\dagger$ The experimental results of these training-free NAS methods are cited from reference (Ha et al., 2024), and the experimental setting of PGD$^7$ is also followed, with the total perturbation scale of 8/255 and the step size of 8/2550 (which is different from PGD$^{20}$ and PGD$^{100}$). Symbol '-' means unmeasured.

As can be observed, the proposed method achieves the second best natural accuracy. Comparing the adversarial accuracy, the proposed method surpasses other peer competitors under simple attacks (e.g., FGSM and PGD). To our surprise, the adversarial accuracy of the proposed method also appears to be the best among all competitors under some stronger attacks (e.g., APGD$_{CE}$ and AA), which demonstrates the effectiveness of the proposed zero-cost proxy for adversarial robustness evaluation. The biggest highlight of the proposed method lies in its search efficiency, which is at least 20 times higher than previous methods that require training, and also about at least 6 times higher than previous training-free methods. Additional experimental results under white-box attacks can be found in Appendix B.1 and B.2.

Table 2: Results of adversarially trained models on CIFAR-10 under transfer-based black-box attacks.

| Source \ Target | DSRNA | RACL | AdvRush | Ours | Cost |
|---|---|---|---|---|---|
| DSRNA | - | 64.96% | 66.86% | 64.40% | 0.4 |
| RCAL | 62.37% | - | 66.37% | 64.77% | 0.5 |
| AdvRush | 62.26% | 64.22% | - | 64.24% | 0.7 |
| **Ours** | **61.24%** | **64.19%** | **66.12%** | - | **0.017** |

### 4.1.3 PERFORMANCE UNDER BLACK-BOX ATTACKS

To explore the performance of the searched architecture under black-box attacks, we perform transfer-based attacks, attacking target models using adversarial examples generated by source models. The results are shown in Table 2. In each column, when our architecture is used as a source model (in bold), the accuracy of its target model is always the lowest, indicating that our architecture generates the strongest transfer-based black-box attacks. When considering each model pair, we can further compare the adversarial robustness between every two models. For example, when considering the model pair Ours $\leftrightarrow$ DSRNA, Ours $\rightarrow$ DSRNA achieves the attack success rate (i.e., 100% - adversarial accuracy) of 38.76%, while DSRNA $\rightarrow$ Ours achieves the attack success rate of 35.60%, indicating that our architecture has stronger adversarial robustness under the black-box attack. Similarly, our architecture has stronger adversarial robustness to transfer-based black-box attacks than RACL. Additional experimental results under black-box settings can be found in Appendix B.3.

### 4.1.4 TRANSFERABILITY TO OTHER DATASETS

The architecture searched on CIFAR-10 is transferred to SVHN and Tiny-ImageNet-200 to show its transferability. The results are shown in Table 3. When transferred to SVHN, our architecture performs best under all metrics. When transferred to Tiny-ImageNet-200, it demonstrates a significant improvement. Compared with ResNet-18, our architecture is far ahead with 17.66% higher natural accuracy, 16.73% higher FGSM accu-

Table 3: Evaluation results of adversarially trained models on SVHN and Tiny-ImageNet-200.

| Datasets | Model | Natural Acc. | FGSM | PGD$^{20}$ |
|---|---|---|---|---|
| SVHN | ResNet-18 | 92.06% | 88.73% | 69.51% |
| | PDARTS | 95.10% | 93.01% | 89.58% |
| | **Ours** | **95.79%** | **95.14%** | **91.64%** |
| Tiny-ImageNet-200 | ResNet-18 | 36.26% | 16.08% | 13.94% |
| | PDARTS | 45.94% | 24.36% | **22.74%** |
| | **Ours** | **53.92%** | **32.81%** | 15.72% |

racy, and 1.78% higher PGD$^{20}$ accuracy. Compared with PDARTS, the natural and FGSM accuracy of our architecture are also significantly higher, though its PGD$^{20}$ accuracy is lower. Additional experimental results and analysis regarding the transferability can be found in Appendix B.4.

### 4.1.5 PERFORMANCE ON THE LARGER-SCALE DATASET

In order to evaluate the proposed zero-cost proxy on the larger-scale dataset, we have performed experiments on ImageNet (Deng et al., 2009) and compared the proposed zero-cost proxy with state-of-the-art robust zero-cost proxy (i.e., CRoZe (Ha et al., 2024)) and robust NAS methods (i.e., RACL (Dong et al., 2020) and Ad-vRush (Mok et al., 2021)). To perform the evaluation, we train the architec-

Table 4: Experimental results of the proposed zero-cost proxy and state of the arts on ImageNet.

| Methods | With Training? | Natural Acc. (%) | FGSM (%) | PGD$^{20}$ (%) |
|---|---|---|---|---|
| RACL (Dong et al., 2020) | ✓ | 51.59% | 18.15% | 10.49% |
| AdvRush (Mok et al., 2021) | ✓ | 51.54% | 18.42% | 10.74% |
| CRoZe (Ha et al., 2024) | ✗ | 49.95% | 16.54% | 9.67% |
| Ours | ✗ | **52.71%** | **19.88%** | **11.96%** |

tures derived by both proxies for 30 epochs, with the learning rate decayed by the factor of 0.1 at 20-th and 25-th epochs. Besides, because the vanilla adversarial training on ImageNet will cost thousands of GPU days, we used the fast adversarial training technique (Wong et al., 2020) for acceleration. The experimental results are presented in Table 4. It is clearly observed that the proposed zero-cost proxy demonstrates the best performance in terms of natural accuracy and accuracy under adversarial attacks. The results indicate that the proposed zero-cost proxy is still effective on larger-scale dataset, further demonstrating the effectiveness of the proposed zero-cost proxy.

### 4.1.6 PERFORMANCE IN LARGER SEARCH SPACE

In addition to the results in DARTS search space (Liu et al., 2019), we also perform experiments in WideResNet (WRN)-like search space (Zagoruyko, 2016; Li et al., 2021), to show the effectiveness of the proposed zero-cost proxy in the more complex search space. The results are presented in Table 5.

As can be seen from Table 5, the proposed zero-cost proxy can still achieve the best performance in the WRN-like search space, along with the lowest search cost. Although the search space becomes larger and more complex, the search cost of our proxy just shows slightly increment compared with that in DARTS search space (from 0.017 GPU days to 0.019 GPU days). This phenomenon further demonstrates the efficiency of the proposed zero-cost proxy. More experimental results about the computational cost and the scalability can be found in Appendix B.5 and B.6, respectively.

### 4.1.7 Ablation Study of the Proposed Zero-cost Proxy

To show the individual role of the proposed zero-cost proxy in robust NAS, we compare it with the mainstream weight-sharing technique adopted by peer competitors. Specifically, we adversarially train a super-network of the DARTS search space, evaluate randomly sampled architectures using the shared weights, and train the best architecture from scratch. The results are shown in Table 6.

Table 5: Experimental results in WRN-like search space. The derived architectures are adversarially trained on CIFAR-10.

| Models | Natural Acc. (%) | FGSM (%) | PGD$^{20}$ (%) | Search Cost (GPU Days) |
|---|---|---|---|---|
| NADAR (Li et al., 2021) | 86.23% | 60.46% | 53.43% | 0.5 |
| **Ours** | **87.34%** | **61.93%** | **53.89%** | **0.019** |

The total cost (cost of training super-network + search cost) of the proposed zero-cost proxy is 500 times less than the weight-sharing technique. Meanwhile, given that both methods have their own advantages in terms of natural accuracy and adversarial accuracy, we use Harmonic Robustness Score (HRS) (Devaguptapu et al., 2021) for further comparison. HRS is

Table 6: Ablation study of the proposed zero-cost proxy. The cost is measured in GPU Days.

| Performance Estimation | Cost of Training Super-network | Search Cost | Natural Acc. | PGD$^{20}$ | HRS |
|---|---|---|---|---|---|
| Weight-sharing | 6.3 | 2.2 | **84.34%** | 50.79% | 63.40 |
| The proposed zero-cost proxy | 0.0 | 0.017 | 83.67% | **52.00%** | **64.14** |

a recently-introduced metric that considers both natural accuracy and adversarial accuracy. The higher the HRS, the better the architecture. The proposed zero-cost proxy gets a higher HRS score, which means the architecture searched by the proposed zero-cost proxy is better.

### 4.1.8 Architecture Ingredients of Adversarially Robust Neural Networks

To analyze the architecture ingredients of adversarially robust networks, we randomly sample 1,000 architectures from the DARTS search space and obtain the top 100 architectures according to their scores. The statistical data is shown in Figure 1. On the whole, most architectures in the top 100 tend to employ a large number of learnable primitives, but few primitives without learnable parameters. Under this overall impression, we further analyze each trend line in detail.

When analyzing the trend line corresponding to skip connections, we learn that most top 100 architectures are without skip connections, and the number of architectures decreases dramatically as the number of skip connections increases. This is a special case in which the proposed proxy may misjudge the adversarial robustness. Commonly, the architectures without skip connections will not perform well in terms of adversarial robustness, because they may meet with training difficulties such as gradient vanishing (Glorot & Bengio, 2010). The proposed method may misjudge such

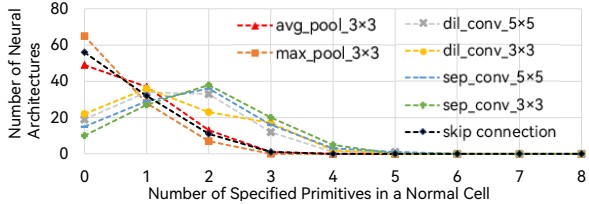

Figure 1: Analysis on the architecture ingredients of Top-100 neural architectures.

neural architectures as good ones, because it evaluates the adversarial robustness of neural networks through their weights at initialization but ignores the effect of skip connections.

Similarly, both $3 \times 3$ max pooling and average pooling have similar tendencies to skip connections, indicating that they have little effect on the adversarial robustness. Differently, as the horizontal value gets large, the trend lines corresponding to learnable primitives first rise and then fall, indicating that all types of learnable primitives contribute to adversarial robustness. But the trend line of $3 \times 3$ dilated convolutions falls earlier, indicating that robust architectures prefer $5 \times 5$ dilated convolutions, $3 \times 3$ separable convolutions, and $5 \times 5$ separable convolutions more than $3 \times 3$ dilated convolutions. Furthermore, more analysis of architecture ingredients are placed in Appendix B.7.

## 4.2 RQ2: SUPERIORITY IN TERMS OF EVALUATING ADVERSARIAL ROBUSTNESS

### 4.2.1 EXPERIMENTAL SETTINGS

**Datasets** NAS-Bench-201-R (Jung et al., 2023): NAS-Bench-201-R is the first to test all pretrained NAS-Bench-201 models (containing 6,466 unique architectures) under various adversarial attacks, i.e., FGSM, PGD, APGD, and Square, with various attack parameters. It is effective to evaluate zero-cost proxies under weak attacks. However, it cannot work when attacks are strong, because their models are not adversarially trained, resulting in near zero accuracies when suffering strong adversarial attacks. Such data is not informative.

Tiny-RobustBench: To effectively evaluate zero-cost proxies when the attacks are strong, we create a dataset called Tiny-RobustBench, which maps 223 adversarially trained networks to their adversarial accuracy. The neural architectures are sampled from the DARTS search space (Liu et al., 2019), which is the most popular one in the robust NAS community (Guo et al., 2020; Mok et al., 2021). The adversarial training method also follows the advanced settings in the robust NAS community (Mok et al., 2021; Ou et al., 2024). Please note that acquiring the dataset is expensive, because the adversarial training method with the advanced settings takes more than 10 times as long as the standard training, which may be an obstacle to relevant researchers. We believe that creating such a dataset is also one of our contributions, and the dataset is open-sourced for further research. More details of Tiny-RobustBench can be found in Appendix C.1.

**Peer Competitors** We take state-of-the-art zero-cost proxies as peer competitors, i.e., grad_norm (Chen et al., 2018), snip (Lee et al., 2018), grasp (Wang et al., 2020), synflow (Tanaka et al., 2020), fisher (Turner et al., 2019), jacob_cov (Mellor et al., 2021), and MGM (Xu et al., 2021).

**Parameter Settings** Our zero-cost proxy has only five parameters in Eq. (4). Following the settings in MGM (Xu et al., 2021), the number of layers $M$ and the number of data samples $N$ are set to 11 and 25, respectively. $h$ is set to 50. $t$ that represents the prediction epoch is set to $5 \times 10^6$, which plays the role of balancing the two parts divided by the multiple sign in Eq. (4) in practice. The batch size is set to 8. The parameter studies are presented in Appendix C.2.

### 4.2.2 CORRELATION TO VARIOUS ADVERSARIAL ACCURACY ON NAS-BENCH-201-R

We calculate Kendall's Tau correlation of our zero-cost proxy on NAS-Bench-201-R. The results are shown in Figure 2. The proposed zero-cost proxy is positively correlated to the adversarial accuracy of these four attacks ($\tau = 0.38$) when the attacks are weak ($e = 0.1/255$). When the attacks get stronger ($e$ gets larger), $\tau$ gets larger under the FGSM attack, indicating the proposed zero-cost proxy makes more accurate evaluations under stronger FGSM attacks. However, the correlations become slightly weaker under stronger PGD attacks, and even vanish under Square attacks and APGD attacks.

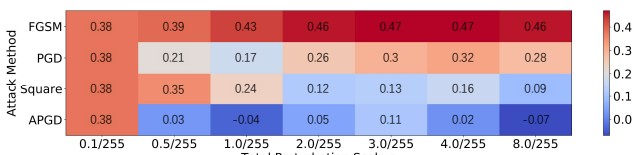

Figure 2: Kendall's Tau correlation coefficient of the proposed zero-cost proxy under various attacks with various parameters in the NAS-Bench-201-R Dataset. Symbol $e$ represents the total perturbation scale of these attacks.

This phenomenon is not in line with practice. According to recent research (Mok et al., 2021; Ou et al., 2024) as well as our practice, networks performing well under one attack tend to keep

outstanding under other attacks (though not absolute). It is curious that the proposed zero-cost proxy shows positive correlations to FGSM attacks but no correlation to Square and APGD attacks. We conjecture this is because the networks in the NAS-Bench-201-R dataset are not adversarially trained, so they generate invalid predictions when attacks get stronger. For example, under PGD attacks with $e = 8.0/255$, the adversarial accuracy of these networks is near zero, and the situation worsens under Square and APGD. Consequently, we recommend testing correlations with adversarially trained networks using the proposed Tiny-RobustBench dataset, which is also in line with our practical needs.

### 4.2.3 Correlation to Adversarial Accuracy of Adversarially Trained Networks in Tiny-RobustBench Dataset

To show the effectiveness of the proposed zero-cost proxy in evaluating adversarially trained networks, we evaluate all networks in the Tiny-RobustBench dataset using the proposed proxy and obtain their score rankings. The results are visualized in Figure 3. Figure 3a is a scatter plot, where each point represents a network corresponding to its score ranking and its adversarial accuracy, and the solid blue line is the result of linear regression. From the solid blue line, we can see that the larger the score ranking (representing a lower score), the lower the adversarial accuracy. Figure 3b is a histogram, where 223 networks are classified into four groups according to their score rankings, and each bar represents their average adversarial accuracy.

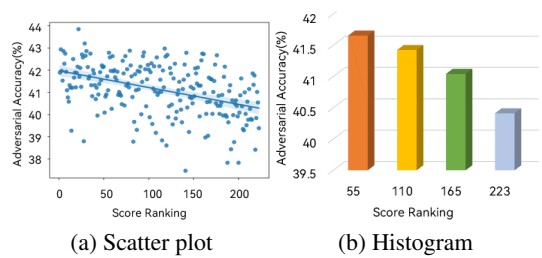

(a) Scatter plot   (b) Histogram

Figure 3: Visualization of the correlation between the score rankings and adversarial accuracy.

sents their average adversarial accuracy. This figure shows that the average adversarial accuracy decreases when the score becomes lower. From these two figures, we come to the same conclusion that the proposed zero-cost proxy is positively correlated to adversarial robustness, which is in line with the theory during the design.

### 4.2.4 Comparison with Previous Zero-cost Proxies

To show the superiority of our zero-cost proxy, we compare it with chosen peer competitors, and the results are visualized in Figure 4. Our proxy reaches the highest Kendall's Tau correlation coefficients, indicating the proposed proxy is most correlated to adversarial robustness. Interestingly, MGM was originally designed to be positively correlated to natural accuracy, but now shows the strongest negative correlation to adversarial accuracy. This is because MGM is designed according to the upper bound of the natural loss, while the natural loss always conflicts with the adversarial loss (Tsipras et al., 2018). Furthermore, more comparisons between the proposed zero-cost proxy and MGM are presented in Appendix C.3.

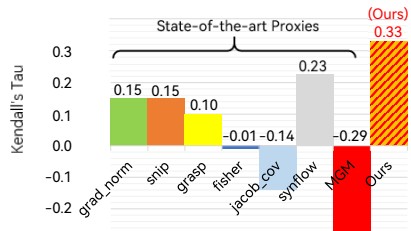

Figure 4: Kendall's Tau correlation coefficient of zero-cost proxies in the Tiny-RobustBench dataset.

## 5 Conclusion

In this work, we recognize that existing techniques for adversarial robustness evaluation are computationally expensive. To address the problem, we design a zero-cost proxy, evaluating the adversarial robustness of DNNs according to their initialized weights without training. The proposed zero-cost proxy is shown to be effective to NAS, descending the search cost dozens of times less than state-of-the-art robust NAS methods. Meanwhile, the proposed zero-cost proxy shows the strongest correlation to adversarial robustness among existing zero-cost proxies. In addition, we also open-source a dataset called Tiny-RobustBench to promote the development of the community.

ACKNOWLEDGMENTS

This work was supported by National Natural Science Foundation of China under Grant 62276175 and Innovative Research Group Program of Natural Science Foundation of Sichuan Province under Grant 2024NSFTD0035.

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

APPENDIX

The appendix contains additional detailed information not covered in the main article. The organization of the appendix is presented as follows:

- Appendix A: This section contains additional theoretical analysis of the proposed zero-cost proxy.
- Appendix B: This section contains additional experimental results for **RQ1**.
- Appendix C: This section contains additional experimental results for **RQ2**.
- Appendix D: Some visualizations of the neural architectures derived by the NAS method along with the proposed zero-cost proxy.

## A  ADDITIONAL THEORETICAL ANALYSIS OF THE PROPOSED ZERO-COST PROXY

### A.1  ASSUMPTIONS OF THE PROPOSED ZERO-COST PROXY

In this section, we discuss the assumption in terms of the infinite-width DNN parameters. In particular, when transferring Eq. (1) to Eq. (3), the assumption of infinite-width DNN parameters is valid. As evidenced by the previous literature Jacot et al. (2018), in the infinite-width limit, the NTK becomes deterministic at initialization and stays constant during training. Consequently, when replacing $\Theta_{\theta_t}$ in Eq. (1) with $\Theta_{\theta_0}$ in Eq. (3), because of the invariance of NTK in the infinite-width limit, these two NTKs can be directly replaced by each other. Therefore, the transformation from Eq. (1) to Eq. (3) keeps valid for the infinite-width DNN parameters.

### A.2  VALIDITY OF APPROXIMATING $-\lambda_{min}(\hat{\Theta}_{\theta_0})$ WITH $\lambda_{min}(\Theta_{\theta_0})$

In this section, we theoretically justify the validity of Approximating $-\lambda_{min}(\hat{\Theta}_{\theta_0})$ with $\lambda_{min}(\Theta_{\theta_0})$. Suppose $x$ and $y$ denote image samples and the corresponding label vector, and $\hat{x}$ denotes the adversarial examples generated based on the input samples $x$. We can obtain the lower bound of $\lambda_{min}(\Theta_{\theta_0})$ based on Eq. (3):

$$\lambda_{min}(\Theta_{\theta_0}) \geq \frac{1}{t} \ln \frac{\|y - f_{\theta_0}(x)\|_2^2}{\|y - f_{\theta_t}(x)\|_2^2}. \tag{9}$$

Similarly, the lower bound of $\lambda_{min}(\hat{\Theta}_{\theta_0})$ can be obtained based on Eq. (5):

$$\lambda_{min}(\hat{\Theta}_{\theta_0}) \geq \frac{1}{t} \ln \frac{\|y - f_{\theta_0}(\hat{x})\|_2^2}{\|y - f_{\theta_t}(\hat{x})\|_2^2}. \tag{10}$$

Based on Eq. (9) and Eq. (10), we can obtain the inequality of $\lambda_{min}(\Theta_{\theta_0})$ and $\lambda_{min}(\hat{\Theta}_{\theta_0})$ by adding the two inequalities together:

$$\lambda_{min}(\Theta_{\theta_0}) + \lambda_{min}(\hat{\Theta}_{\theta_0}) \geq \frac{1}{t} \ln \frac{\|y - f_{\theta_0}(x)\|_2^2 \|y - f_{\theta_0}(\hat{x})\|_2^2}{\|y - f_{\theta_t}(x)\|_2^2 \|y - f_{\theta_t}(\hat{x})\|_2^2}. \tag{11}$$

To form the equivalence of $\lambda_{min}(\Theta_{\theta_0})$ and $\lambda_{min}(\hat{\Theta}_{\theta_0})$, we introduce a margin $\xi \geq 0$, and then the equivalence of $\lambda_{min}(\Theta_{\theta_0})$ and $\lambda_{min}(\hat{\Theta}_{\theta_0})$ can be formed as Eq. (12) based on Eq. (11):

$$\lambda_{min}(\hat{\Theta}_{\theta_0}) = -\lambda_{min}(\Theta_{\theta_0}) + \frac{1}{t} \ln \frac{\|y - f_{\theta_0}(x)\|_2^2 \|y - f_{\theta_0}(\hat{x})\|_2^2}{\|y - f_{\theta_t}(x)\|_2^2 \|y - f_{\theta_t}(\hat{x})\|_2^2} + \xi. \tag{12}$$

According to the theoretical findings of Mok et al. (2022), $\lambda_{min}(\Theta_{\theta_0})$ and $\lambda_{min}(\hat{\Theta}_{\theta_0})$ satisfy the upper bound shown in Eq. (13) and Eq. (14):

$$\lambda_{min}(\Theta_{\theta_0}) \leq \sqrt{\sum_k |\lambda_k(\Theta_{\theta_0})|^2}, \tag{13}$$

$$\lambda_{min}(\hat{\Theta}_{\theta_0}) \leq \sqrt{\sum_k |\lambda_k(\hat{\Theta}_{\theta_0})|^2}, \tag{14}$$

where $k$ denotes the number of eigenvalues. Meanwhile, because the loss of the network with trained weights $\theta_t$ is often smaller than that of the network with randomly initialized weights $\theta_0$ because of the training, the losses $\|y - f_{\theta_0}(x)\|_2^2$ and $\|y - f_{\theta_0}(\hat{x})\|_2^2$ are often larger than $\|y - f_{\theta_t}(x)\|_2^2$ and $\|y - f_{\theta_t}(\hat{x})\|_2^2$. Consequently, $\frac{1}{t} \ln \frac{\|y - f_{\theta_0}(x)\|_2^2 \|y - f_{\theta_0}(\hat{x})\|_2^2}{\|y - f_{\theta_t}(x)\|_2^2 \|y - f_{\theta_t}(\hat{x})\|_2^2}$ is often larger than 0. Therefore, the upper bound of $\xi$ can be formulated as Eq. (15):

$$\xi \leq \sqrt{\sum_k |\lambda_k(\Theta_{\theta_0})|^2} + \sqrt{\sum_k |\lambda_k(\hat{\Theta}_{\theta_0})|^2}. \tag{15}$$

Based on Eq. (12) and Eq. (15), we can conclude that there exists a margin $\xi$ satisfying $0 \leq \xi \leq \sqrt{\sum_k |\lambda_k(\Theta_{\theta_0})|^2} + \sqrt{\sum_k |\lambda_k(\hat{\Theta}_{\theta_0})|^2}$ such that $\lambda_{min}(\hat{\Theta}_{\theta_0})$ and $\lambda_{min}(\Theta_{\theta_0})$ negatively correlated. Therefore, the validity of approximating $-\lambda_{min}(\hat{\Theta}_{\theta_0})$ with $\lambda_{min}(\Theta_{\theta_0})$ is justified.

# B ADDITIONAL EXPERIMENTAL RESULTS FOR RQ1

## B.1 COMPARISONS UNDER WHITE-BOX ATTACKS ON CIFAR-100 AND IMAGENET

In addition to the comparisons on CIFAR-10, we have also performed comparisons on CIFAR-100 and ImageNet. Please note that part of the comparison on ImageNet has been presented in Table 4. In this section, additional results on ImageNet are presented under more adversarial attacks (i.e., PGD[100] and AA) in terms of the models chosen in Table 1, in order to achieve a more comprehensive comparison. The experimental results are shown in Table 7.

Table 7: Comparisons with state of the arts on CIFAR-100 and ImageNet under the white-box attacks.

| Dataset | Model | Training-Free? | Natural Acc. | FGSM | PGD[20] | PGD[100] | AA |
|---------|-------|----------------|--------------|------|---------|----------|-----|
| CIFAR-100 | ResNet-18 (He et al., 2016) | × | 55.12% | 25.65% | 21.08% | 19.98% | 18.02% |
| | DenseNet-121 (Huang et al., 2017) | × | **61.71%** | 34.28% | 27.30% | 27.07% | 24.55% |
| | DARTS (Liu et al., 2019) | × | 59.14% | 30.35% | 25.66% | 25.40% | 22.65% |
| | PDARTS (Chen et al., 2019) | × | 58.41% | 34.81% | 29.11% | 28.87% | 24.07% |
| | RACL (Dong et al., 2020) | × | 59.18% | 32.04% | 26.61% | 26.20% | 22.92% |
| | DSRNA (Hosseini et al., 2021) | × | 57.44% | 35.03% | 28.11% | 27.97% | 25.20% |
| | WsrNet (Cheng et al., 2023) | × | 57.81% | 28.08% | 23.27% | 23.01% | 21.57% |
| | GradNorm (Chen et al., 2018) | ✓ | 58.66% | 32.87% | 28.33% | 28.05% | 25.58% |
| | SynFlow (Tanaka et al., 2020) | ✓ | 55.66% | 30.08% | 25.24% | 24.90% | 22.46% |
| | CroZe (Ha et al., 2024) | ✓ | 59.23% | 30.31% | 26.16% | 26.02% | 22.82% |
| | **Ours** | ✓ | 59.39% | **35.73%** | **32.10%** | **31.88%** | **29.95%** |
| ImageNet | ResNet-18 (He et al., 2016) | × | 47.38% | 17.88% | 8.88% | 8.37% | 7.91% |
| | DenseNet-121 (Huang et al., 2017) | × | 44.13% | 12.51% | 3.74% | 3.14% | 3.72% |
| | DARTS (Liu et al., 2019) | × | 50.58% | 17.45% | 10.07% | 9.44% | 8.53% |
| | PDARTS (Chen et al., 2019) | × | 51.56% | 18.10% | 10.18% | 9.40% | 8.77% |
| | RACL (Dong et al., 2020) | × | 51.59% | 18.15% | 10.49% | 9.82% | 8.99% |
| | DSRNA (Hosseini et al., 2021) | × | 43.32% | 13.04% | 7.88% | 7.49% | 6.47% |
| | WsrNet (Cheng et al., 2023) | × | 50.93% | 17.58% | 10.15% | 9.48% | 8.77% |
| | GradNorm (Chen et al., 2018) | ✓ | 51.34% | 18.30% | 10.71% | 9.90% | 9.25% |
| | SynFlow (Tanaka et al., 2020) | ✓ | 51.47% | 18.90% | 10.73% | 9.93% | 9.42% |
| | CroZe (Ha et al., 2024) | ✓ | 49.95% | 16.54% | 9.67% | 9.10% | 8.36% |
| | **Ours** | ✓ | **52.71%** | **19.88%** | **11.96%** | **10.94%** | **10.38%** |

As can be observed from Table 7, the proposed zero-cost proxy achieves the second-best natural accuracy on CIFAR-100. Meanwhile, the proposed zero-cost proxy demonstrate the state-of-the-art under all the adversarial attacks chosen. Furthermore, the proposed zero-cost proxy still achieves the highest natural accuracy and adversarial robustness on ImageNet comparing with the models chosen in Table 1. In summary, the experimental results in Table 7 show the effectiveness of the proposed zero-cost proxy on multiple datasets besides CIFAR-10.

## B.2 MORE STATISTICAL RESULTS OF THE PROPOSED ZERO-COST PROXY

In order to evaluate the performance of the proposed zero-cost proxy more comprehensively, we have conduct experiments on CIFAR-10 to obtain the statistical results. Specifically, we have run the

proposed zero-cost proxy twice on CIFAR-10, and then adversarially trained the derived architectures following the settings presented in Section 4.1.1. Then, the proposed zero-cost proxy is compared with the widely-compared zero-cost proxy (i.e., NASWOT (Mellor et al., 2021) and NASI (Shu et al., 2022a)), and the experimental results are reported in the "mean value $\pm$ standard deviation" format in Table 8.

Table 8: Statistical results of the proposed zero-cost proxy and widely-compared zero-cost proxies on CIFAR-10.

| Model | Natural Acc. (%) | PGD$^{20}$ (%) |
|---|---|---|
| NASWOT (Mellor et al., 2021) | $80.89\% \pm 2.33\%$ | $51.53\% \pm 0.89\%$ |
| NASI (Shu et al., 2022a) | $78.75\% \pm 0.18\%$ | $50.07\% \pm 0.26\%$ |
| **Ours** | **$82.90\% \pm 0.77\%$** | **$51.98\% \pm 0.02\%$** |

As shown in Table 8, the proposed zero-cost proxy still achieves highest mean values in terms of the natural accuracy and adversarial robustness. The results demonstrate that the proposed zero-cost proxy has better ability than NASWOT and NASI on discovering neural architectures with better adversarial robustness. In addition, please note that the results shown in Table 1 are reported based on the single run. This is because there is a convention in the robust NAS community to report the best architecture for comparison (Guo et al., 2020; Mok et al., 2021). But we still present the statistical results in this section for a more complete evaluation of the proposed zero-cost proxy.

## B.3 COMPARISONS UNDER BLACK-BOX ATTACKS ON CIFAR-100 AND IMAGENET

In order to make the comparison under black-box attacks more comprehensive, the experimental results are carried out on multiple datasets (i.e., CIFAR-100 and ImageNet) in terms of the baselines chosen in Table 2. Besides, two more recent methods (i.e., WsrNet (Cheng et al., 2023) and CroZe (Ha et al., 2024)) are also contained in the experiments. The experimental results are shown in Table 9.

Table 9: Comparisons with state of the arts on CIFAR-100 and ImageNet under the black-box attack.

| Dataset | Source \ Target | DSRNA | RACL | AdvRush | WsrNet | CroZe | Ours |
|---|---|---|---|---|---|---|---|
| CIFAR-100 | DSRNA (Hosseini et al., 2021) | - | 41.74% | 45.04% | 42.73% | 43.13% | 45.72% |
| | RCAL (Dong et al., 2020) | 42.90% | - | 44.04% | 42.84% | 44.33% | 46.87% |
| | AdvRush (Mok et al., 2021) | 41.83% | **39.60%** | - | 41.66% | 43.30% | 45.53% |
| | WsrNet (Cheng et al., 2023) | 44.66% | 43.30% | 47.13% | - | 46.38% | 47.68% |
| | CroZe (Ha et al., 2024) | 41.98% | 41.68% | 45.06% | 43.34% | - | 46.15% |
| | **Ours** | **39.90%** | 40.43% | **43.08%** | **40.89%** | **41.76%** | - |
| ImageNet | DSRNA (Hosseini et al., 2021) | - | 31.50% | 31.74% | 29.38% | 27.97% | 36.69% |
| | RCAL (Dong et al., 2020) | 23.77% | - | 27.48% | 27.49% | 26.51% | 32.72% |
| | AdvRush (Mok et al., 2021) | 23.65% | 27.22% | - | 27.37% | 26.23% | 32.54% |
| | WsrNet (Cheng et al., 2023) | 22.45% | 27.97% | 28.13% | - | 25.75% | 33.40% |
| | CroZe (Ha et al., 2024) | 22.41% | 28.14% | 28.17% | 26.94% | - | 33.71% |
| | **Ours** | **21.72%** | **24.43%** | **24.49%** | **24.61%** | **23.82%** | - |

As shown in Table 9, our model achieves the highest attack success rate (100% - the test accuracy, highlighted in bold) for all the target models except RACL on CIFAR-100. Meanwhile, our model also achieves the highest attack success rate for all the models on ImageNet. Furthermore, when our model is set as the target model, it demonstrates the best adversarial robustness among all the baselines chosen. In conclusion, the proposed zero-cost proxy is still effective under the black-box attacks across multiple datasets.

## B.4 ADDITIONAL EXPERIMENTAL RESULTS OF THE TRANSFERABILITY

To further evaluate the transferability of the proposed zero-cost proxy, we present the experimental results of models in Table 1 in terms of the transferability. Specifically, these models are directly transferred and trained on SVHN and Tiny-ImageNet-200 for evaluation. The experimental results are shown in Table 10.

Table 10: Additional results of adversarially trained models on SVHN and Tiny-ImageNet-200.

| Datasets | Model | Natural Acc. | FGSM | PGD[20] |
|---|---|---|---|---|
| SVHN | DenseNet-121 (Huang et al., 2017) | 93.72% | 89.68% | 72.62% |
| | DARTS (Liu et al., 2019) | 94.90% | 90.01% | 77.58% |
| | RobNet-free (Guo et al., 2020) | 92.45% | 89.33% | 85.30% |
| | DSRNA (Hosseini et al., 2021) | 91.58% | 91.27% | 84.94% |
| | WsrNet (Cheng et al., 2023) | 94.97% | 76.67% | 84.20% |
| | GradNorm (Chen et al., 2018) | 95.16% | 92.51% | 90.53% |
| | SynFlow (Tanaka et al., 2020) | 95.52% | 91.53% | 76.96% |
| | CroZe (Ha et al., 2024) | 93.19% | 66.36% | 48.11% |
| | **Ours** | **95.79%** | **95.14%** | **91.64%** |
| Tiny-ImageNet-200 | DenseNet-121 (Huang et al., 2017) | 46.26% | 22.88% | 19.11% |
| | DARTS (Liu et al., 2019) | 45.94% | 24.36% | 21.74% |
| | RobNet-free (Guo et al., 2020) | 44.24% | 25.44% | 23.85% |
| | DSRNA (Hosseini et al., 2021) | 44.42% | 28.52% | **24.32%** |
| | WsrNet (Cheng et al., 2023) | 48.62% | 22.65% | 19.86% |
| | GradNorm (Chen et al., 2018) | 49.17% | 16.02% | 11.35% |
| | SynFlow (Tanaka et al., 2020) | 50.96% | 12.80% | 8.13% |
| | CroZe (Ha et al., 2024) | 48.06% | 10.17% | 6.08% |
| | **Ours** | **53.92%** | **32.81%** | 15.72% |

As can be seen from Table 10, when comparing with the models beyond ResNet-18 and PDARTS in Table 3, our architecture still achieves decent natural accuracy and adversarial robustness, demonstrating superior transferability. Furthermore, we have also provided justification for the transferability grounded in our theoretical contributions.

**Justification for the transferability**  The transferability of the proposed zero-cost proxy can be attribute to the design of the zero-cost proxy. In particular, when evaluating the adversarial robustness $R$ of the architecture, the input samples $x$ are not specified. Meanwhile, the initial weights $\theta_0$ of the network is randomly determined, without any specific constraints. Consequently, the proposed zero-cost proxy can find the inherently robust neural architectures, without the dependence on specific data or weights. As a result, the architecture derived on a certain dataset based on the proposed zero-cost proxy can be well transferred to other datasets.

## B.5 MORE COMPARISON OF THE SEARCH COST

In order to evaluate the efficiency of the proposed zero-cost proxy more comprehensively, we carry out more experimental results in terms of the search cost. Specifically, we compare the proposed zero-cost proxy with the state-of-the art robust zero-cost proxy CRoZe (Ha et al., 2024). Besides, the benchmark dataset chosen are CIFAR-10 and ImageNet, and the search spaces are DARTS space (Liu et al., 2019) and WRN space (Zagoruyko, 2016; Li et al., 2021). The experimental results are presented in Table 11.

Table 11: The comparisons in terms of the search cost on different datasets and search spaces. The search cost is measured in GPU days.

| Datasets | Search Spaces | Methods | Cost |
|---|---|---|---|
| CIFAR-10 | DARTS Space | CRoZe (Ha et al., 2024) | 0.2 |
| | DARTS Space | Ours | 0.017 |
| | WRN Space | Ours | 0.019 |
| ImageNet | DARTS Space | Ours | 0.036 |

As can be seen from Table 11, when the search space changed from DARTS space to WRN space, the search cost of the proposed zero-cost proxy increases slightly (from 0.017 GPU days to 0.019

GPU days). Meanwhile, when the datasets change from CIFAR-10 to ImageNet, the proposed zero-cost proxy demonstrates the similar behavior. Besides, no matter which dataset or search space is adopted, the proposed zero-cost proxy always achieves higher efficiency than CRoZe (Ha et al., 2024), demonstrating the efficiency of the proposed zero-cost proxy. This phenomenon can be mainly attributed to the fact that CRoZe needs to generate adversarial examples to achieve the evaluation. In contrast, the proposed zero-cost proxy does not need to generate adversarial examples in evaluation, thus the efficiency is achieved.

## B.6 SCALABILITY TO THE REINFORCEMENT LEARNING

In order to demonstrate the scalability of the proposed zero-cost proxy to other kinds of NAS methods, we integrate our proxy into the reinforcement learning (RL)-based NAS method (i.e., NASNet (Zoph et al., 2018)) to perform experiments. Specifically, the proposed zero-cost proxy is adopted to replace the training-based performance validation in RL-based NAS, and the reward is determined according to the output of the proxy. Then, the derived architectures are forward to adversarial training for evaluation. The experimental results are shown in Table 12.

Table 12: Experimental results in terms of the scalability to the RL-based NAS method. In this set of experiments, the proposed zero-cost proxy is integrated into the NASNet (Zoph et al., 2018) to search for robust neural architectures. The search cost is measured in GPU days.

| Model | Natural Acc. (%) | FGSM (%) | PGD[20] (%) | Cost |
|---|---|---|---|---|
| NASNet (Zoph et al., 2018) | 80.61% | 54.19% | 50.25% | 2,000 |
| NASNet + Our Proxy | **87.35%** | **62.21%** | **53.23%** | **0.027** |

As can be seen from Table 12, the introduction of the proposed zero-cost proxy brings improvements in terms of both natural accuracy and adversarial robustness. Meanwhile, the search cost is significantly reduced (from 2,000 GPU days to 0.027 GPU days). This is because the proposed zero-cost proxy replaces the costly architecture training process in the performance evaluation. In summary, the experimental results demonstrate the effectiveness of the proposed zero-cost proxy in the RL-based NAS method, thus the scalability is proved.

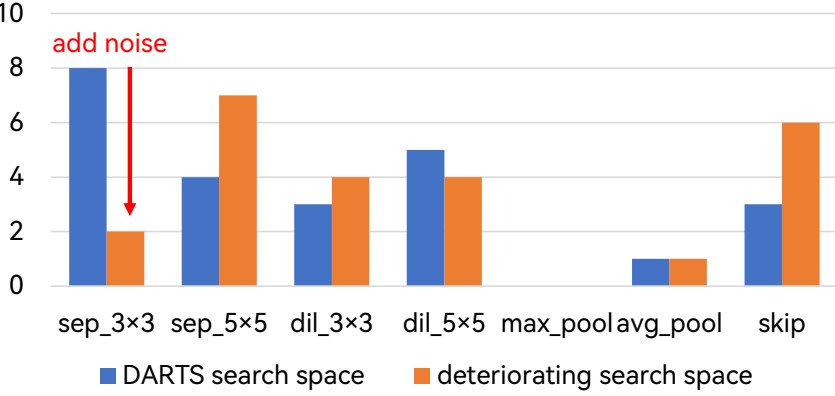

Figure 5: Appearing times of each option in a deteriorating search space.

## B.7 ABILITY TO IDENTIFY "BAD" ARCHITECTURE OPTIONS

So far, all experiments are conducted on the DARTS search space for its effectiveness. We are curious if the proposed method can perform well in a deteriorating search space. For this purpose, We add noise to $3 \times 3$ separable convolutions (one of the popular options as shown in Section 4.1.8) to make it a bad option that is harmful to the DNN performance (Zela et al., 2020), and repeat the proposed method for three times. The appearing times of each option are shown in Figure 5. As shown in the figure, the appearing times of noisy $3 \times 3$ separable convolutions decrease sharply, indicating that the proposed method has the ability to identify bad architecture options and will not choose them.

## C  ADDITIONAL EXPERIMENTAL RESULTS FOR RQ2

### C.1  MORE DETAILS OF THE CONSTRUCTED TINY-ROBUSTBENCH

In this section, we will present more details about the proposed Tiny-RobustBench dataset. The details mainly comes from two aspects, i.e., the training details of the architectures and the data distribution of the whole dataset. In particular, the training details are presented in Table 13.

Table 13: The training details for the architectures in Tiny-RobustBench.

| Items | Values |
|---|---|
| Total Training Epoch | 105 |
| Initial Learning Rate | 0.1 |
| Learning Rate Decay Policy | Stepped Decent |
| Learning Rate Decent Factor | 0.1 |
| The Index of Epoch for Learning Rate Decent | 99 |
| Momentum | 0.9 |
| Weight Decay | 0.0001 |
| Adversarial Loss | PGD |
| Perturbation Rate | 8/255 |
| Number of Steps | 7 |
| Step Size | 0.01 |

In addition, we also present the data distribution in terms of the architectures in the dataset. In particular, we statistically record both adversarial robustness and natural accuracy of the architectures in Tiny-RobustBench, and the distribution is presented in Figure 6. As can be seen, the distribution is relatively dispersed. Such distribution is beneficial for evaluate the performance of zero-cost proxies, because both the better case and the worse case of adversarial robustness are included.

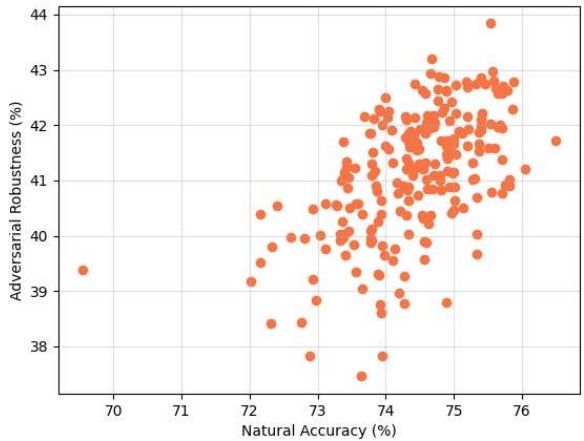

Figure 6: The data distribution of the architectures in Tiny-RobustBench. Specifically, the distribution contains two aspects, i.e., the adversarial robustness and the natural accuracy.

### C.2  HYPER-PARAMETER STUDIES

In order to evaluate different settings of the hyper-parameters $t$ and $h$ in Eq. (4), we have performed hyper-parameter studies for both $t$ and $h$ on the Tiny-RobustBench dataset. Specifically, the hyper-parameter $h$ is fixed to 50, and the hyper-parameter $t$ is set to $5 \times 10^4$, $5 \times 10^5$, $5 \times 10^6$, $5 \times 10^7$, and $5 \times 10^8$ in order to evaluate the impact of $t$ on the performance of the proposed zero-cost proxy. Besides, in order to explore the impact of $h$ on the performance of the proposed zero-cost proxy, the hyper-parameter $t$ is fixed to $5 \times 10^6$, and the hyper-parameter $h$ is set to 0.5, five, 50, $5 \times 10^2$, and $5 \times 10^3$. Please note that $t$ and $h$ are fixed to $5 \times 10^6$ and 50 because this is the default setting

as indicated in Section 4.2.1. The experimental results of the hyper-parameter studies are shown in Tables 14 and 15. In order to evaluate the impact on architecture ranking and architecture search, we report the KTau correlation of the proxy and the accuracy under $PGD^7$ attack of the architecture with the highest rank.

Table 14: Experimental results of the hyper-parameter study for the hyper-parameter $t$. In the experiments, the hyper-parameter $h$ is fixed to 50, and the Tiny-RobustBench dataset is used.

| $t$ $(h = 50)$ | KTau | $PGD^7$ (%) |
|---|---|---|
| $5 \times 10^4$ ($\times 0.01$) | 0.29 | 42.93% |
| $5 \times 10^5$ ($\times 0.1$) | 0.32 | 42.93% |
| $5 \times 10^6$ (Default) | 0.33 | 42.93% |
| $5 \times 10^7$ ($\times 10$) | 0.34 | 42.93% |
| $5 \times 10^8$ ($\times 100$) | 0.26 | 42.93% |

Table 15: Experimental results of the hyper-parameter study for the hyper-parameter $h$. In the experiments, the hyper-parameter $t$ is fixed to $5 \times 10^6$, and the Tiny-RobustBench dataset is used.

| $h$ $(t = 5 \times 10^6)$ | KTau | $PGD^7$ (%) |
|---|---|---|
| 0.5 ($\times 0.01$) | 0.33 | 42.93% |
| 5 ($\times 0.1$) | 0.33 | 41.90% |
| 50 (Default) | 0.33 | 42.93% |
| $5 \times 10^2$ ($\times 10$) | 0.33 | 42.93% |
| $5 \times 10^3$ ($\times 100$) | 0.33 | 42.93% |

As show in Table 14, when the hyper-parameter $t$ decreases to $5 \times 10^4$ or increases to $5 \times 10^6$, the KTau correlation suffers a significant drop. In contrast, the KTau correlation almost keeps unchanged when $t$ decreases to $5 \times 10^5$ or increases to $5 \times 10^6$. Based on these facts, we can conclude that the architecture ranking performance of the zero-cost-proxy will be negatively affected when the hyper-parameter $t$ is too large or too small. Moreover, the architecture with the highest rank keeps the same when the hyper-parameter $t$ is changing, indicating the architecture search is not sensitive to the hyper-parameter $t$.

Furthermore, the results of the hyper-parameter study for $h$ are presented in Table 15. As can be seen, the KTau correlation remains the same when $h$ is changed. Meanwhile, the architecture with the highest rank also keeps unchanged except when $h$ is set to five. Giving all these facts, we can come to the conclusion that the performance of the proposed zero-cost proxy is not sensitive to the hyper-parameter $h$.

In summary, the stability of the proposed zero-cost proxy is demonstrated based on the experimental results shown in Tables 14 and 15, because the performance of the proposed zero-cost proxy is almost not affected when both hyper-parameters change in a large range.

### C.3 FURTHER COMPARISONS WITH MGM PROXY

As shown in Figure 4 in Section 4.2.4, the MGM proxy (Xu et al., 2021) demonstrates strong negative correlation with the adversarial robustness of architectures. Motivated by this fact, we are curious about whether the negative version of the MGM proxy is effective to search for robust architectures. Therefore, we directly use the negative version of MGM proxy to search for architectures in DARTS search space, and then the derived architecture is adversarially trained for evaluation. The experimental results is presented in Table 16, where the natural accuracy and the accuracy under $PGD^{20}$ adversarial attack are reported.

It is clear shown in Table 16 that the proposed zero-cost proxy outperforms the negative version of MGM. Specifically, the proposed zero-cost proxy achieves the improvement of +3.23 and +2.93 for natural accuracy and the accuracy under $PGD^{20}$ adversarial attack, respectively. Therefore, the superiority of the proposed zero-cost proxy is further demonstrated.

Table 16: Comparisons between the negative version of the MGM proxy (Xu et al., 2021) and the proposed zero-cost proxy. The architectures are searched in DARTS search space and adversarially trained on CIFAR-10.

| Model | Natural Acc. (%) | PGD$^{20}$ (%) |
|---|---|---|
| Negative MGM (Xu et al., 2021) | 80.44% | 49.07% |
| Ours | **83.67%** | **52.00%** |

## D ARCHITECTURE VISUALIZATION

The architectures searched by the proposed zero-cost proxy are presented in Figures 7 and 8. Specifically, cells are searched in DARTS search space. The normal cells are presented in Figures 7a and 8a, and the reduction cell are presented in Figures 7b and 8b.

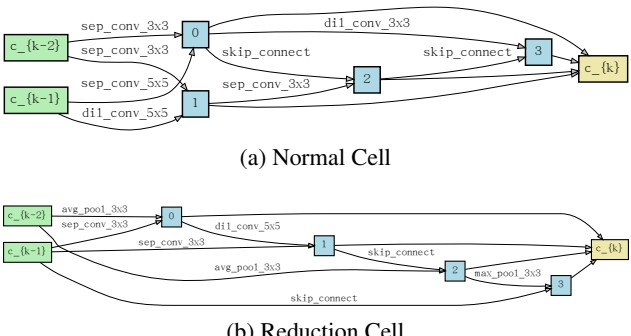

(a) Normal Cell

(b) Reduction Cell

Figure 7: The normal cell and the reduction cell searched by the proposed zero-cost proxy on CIFAR-10.

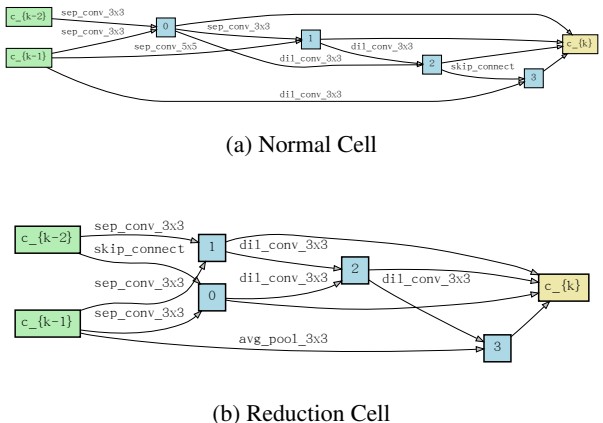

(a) Normal Cell

(b) Reduction Cell

Figure 8: The normal cell and the reduction cell searched by the proposed zero-cost proxy on ImageNet.

