# OpenReview forum: "Zero-cost Proxy for Adversarial Robustness Evaluation"
_ICLR.cc/2025/Conference — ICLR 2025 Poster_

### Official Review · Reviewer_kdEp · 2024-11-01

**Soundness:** 3
**Presentation:** 2
**Contribution:** 3
**Rating:** 6
**Confidence:** 3

**Summary:**

This paper introduces a zero-shot proxy applied directly to the initial weights of a deep neural network (DNN), serving as an upper bound for adversarial loss. This approach enhances the adversarial robustness of the network without requiring adversarial examples. Theoretical justification for the zero-shot proxy is provided based on the Neural Tangent Kernel (NTK) and the input loss landscape. Experimental results demonstrate a 20x speedup over state-of-the-art robust neural architecture search (NAS) methods, with no loss in performance against black-box and white-box attacks.

**Strengths:**

+ The preliminaries on the Neural Tangent Kernel (NTK) and input loss landscape are clearly presented, making the methodology easy to understand. All symbols and equations are well-defined.

+ A robust theoretical justification for the proposed zero-shot proxy is provided by establishing the corresponding loss as an upper bound for adversarial loss.

+ The efficiency of the search process using their technique is clearly demonstrated in the experimental section.

+ The authors introduce a novel dataset called Tiny-RobustBench, which may prove valuable to the research community.

**Weaknesses:**

- In the theoretical analysis section,  $\lambda_{\min}(\hat{\Theta_{\theta_o}})$ is approximated based on empirical observations to $-\lambda_{\min}(\Theta_{\theta_o})$, which weakens the theoretical foundation of the work. I wonder if the authors could establish the relationship between the two NTKs in strict mathematical terms, such as showing that the approximation holds true within a specific margin of error with a certain probability. Currently, the analysis relies on an empirically driven approximation, which undermines its theoretical robustness.
- The performance of the black-box and white-box attacks is demonstrated using a single dataset (CIFAR-10) in Tables 1 and 2. The authors should consider providing performance comparisons across multiple datasets for the chosen baselines in both tables. Additionally, Table 2 should clearly present a comparison of search efficiency.
- I am curious about the rationale for selecting only specific models (ResNet-18 and PDARTS) for comparison in Table 3. Would it be possible to include models from Table 1 to evaluate their transferability in Table 3?
- There is inadequate justification for why their approach exhibits better transferability in Table 3. It would be beneficial to provide justification grounded in their theoretical contributions, particularly highlighting the unique advantages that their proposed techniques offer to enhance transferability.
- The authors should discuss the assumptions made in their analysis and their validity within the context of the study. For instance, when transforming from Equation 1 to Equation 3, it would be helpful to clarify the assumptions involved and how they apply in their setting. Specifically, I am uncertain whether the assumption of infinite-width DNN parameters is valid; if it is not, what implications does that have for this paper? I encourage the authors to include these details, even if they are standard, as doing so would enhance the paper's readability and facilitate the assessment process.

**Questions:**

Please refer to Weaknesses Section

---

> ### Author Response · Authors · 2024-11-22
> **Response to Reviewer kdEp [Part 1/3]**
>
> We are immensely grateful for your effort in reviewing our work, as well as for your constructive suggestions and insightful questions. In the following, we will address each of your suggestions and concerns point-by-point.
>
> >**In the theoretical analysis section, $\lambda_{min}(\hat{\Theta_{\theta_{0}}})$ is approximated based on empirical observations to $-\lambda_{min}(\Theta_{\theta_{0}})$, which weakens the theoretical foundation of the work. I wonder if the authors could establish the relationship between the two NTKs in strict mathematical terms, such as showing that the approximation holds true within a specific margin of error with a certain probability. Currently, the analysis relies on an empirically driven approximation, which undermines its theoretical robustness.**
>
> Thanks for the insightful comment. We establish the relationship between the two NTKs in strict mathematical terms as follows, in order to theoretically justify the validity of the approximation:
>
> Suppose $x$ and $y$ denote image samples and the corresponding label vector, and $\hat{x}$ denotes the adversarial examples generated based on the input samples $x$. We can obtain the lower bound of $\lambda_{min}(\Theta_{\theta_{0}})$ based on $\Vert y-f_{\theta_{t}}(x)\Vert_{2}^{2} \leq \exp(-\lambda_{min}(\Theta_{\theta_{0}})t)\Vert y-f_{\theta_{0}}(x)\Vert_{2}^{2}$:
>
> $\lambda_{min}(\Theta_{\theta_{0}}) \geq \frac{1}{t} \ln \frac{\Vert y - f_{\theta_0}(x) \Vert^2_2}{\Vert y - f_{\theta_t}(x) \Vert^2_2}.$ (1)
>
>
> Similarly, the lower bound of $\lambda_{min}(\hat{\Theta_{\theta_{0}}})$ can be obtained based on $\Vert y-f_{\theta_{t}}(\hat{x})\Vert_{2}^{2}\le \exp(-\lambda_{min}(\hat{\Theta_{\theta_{0}}})t)\Vert y-f_{\theta_{0}}(\hat{x})\Vert_{2}^{2}$:
>
> $\lambda_{min}(\hat{\Theta_{\theta_{0}}}) \geq \frac{1}{t} \ln \frac{\Vert y - f_{\theta_0}(\hat{x}) \Vert^2_2}{\Vert y - f_{\theta_t}(\hat{x}) \Vert^2_2}.$ (2)
>
> Based on Eq.(1) and Eq.(2), we can obtain the inequality of $\lambda_{min}(\Theta_{\theta_{0}})$ and $\lambda_{min}(\hat{\Theta_{\theta_{0}}})$ by adding the two inequalities together:
>
> $\lambda_{min}(\Theta_{\theta_{0}}) + \lambda_{min}(\hat{\Theta_{\theta_{0}}}) \geq \frac{1}{t} \ln \frac{\Vert y - f_{\theta_0}(x) \Vert^2_2 \Vert y - f_{\theta_0}(\hat{x}) \Vert^2_2}{\Vert y - f_{\theta_t}(x) \Vert^2_2 \Vert y - f_{\theta_t}(\hat{x}) \Vert^2_2}.$ (3)
>
> To form the equivalence of $\lambda_{min}(\Theta_{\theta_{0}})$ and $\lambda_{min}(\hat{\Theta_{\theta_{0}}})$, we introduce a margin $\xi \geq 0$, and then the equivalence of $\lambda_{min}(\Theta_{\theta_{0}})$ and $\lambda_{min}(\hat{\Theta_{\theta_{0}}})$ can be formed as Eq.(4) based on Eq.(3):
>
> $\lambda_{min}(\hat{\Theta_{\theta_{0}}}) = -\lambda_{min}(\Theta_{\theta_{0}}) + \frac{1}{t} \ln \frac{\Vert y - f_{\theta_0}(x) \Vert^2_2 \Vert y - f_{\theta_0}(\hat{x}) \Vert^2_2}{\Vert y - f_{\theta_t}(x) \Vert^2_2 \Vert y - f_{\theta_t}(\hat{x}) \Vert^2_2} + \xi.$ (4)
>
> According to the theoretical findings in [1], $\lambda_{min}(\Theta_{\theta_{0}})$ and $\lambda_{min}(\hat{\Theta_{\theta_{0}}})$ satisfy the upper bound shown in Eq.(5) and Eq.(6):
>
> $\lambda_{min}(\Theta_{\theta_{0}}) \leq \sqrt{\sum_k |\lambda_k(\Theta_{\theta_{0}})|^2},$ (5)
>
> $\lambda_{min}(\hat{\Theta_{\theta_{0}}}) \leq \sqrt{\sum_k |\lambda_k(\hat{\Theta_{\theta_{0}}})|^2},$ (6)
>
> where $k$ denotes the number of eigenvalues. Meanwhile, because the loss of the network with trained weights $\theta_t$ is often smaller than that of the network with randomly initialized weights $\theta_0$ because of the training, the losses $\Vert y - f_{\theta_0}(x) \Vert^2_2$ and $\Vert y - f_{\theta_0}(\hat{x}) \Vert^2_2$ are often larger than $\Vert y - f_{\theta_t}(x) \Vert^2_2$ and $\Vert y - f_{\theta_t}(\hat{x}) \Vert^2_2$. Consequently, $\frac{1}{t} \ln \frac{\Vert y - f_{\theta_0}(x) \Vert^2_2 \Vert y - f_{\theta_0}(\hat{x}) \Vert^2_2}{\Vert y - f_{\theta_t}(x) \Vert^2_2 \Vert y - f_{\theta_t}(\hat{x}) \Vert^2_2}$ is often larger than 0. Therefore, the upper bound of $\xi$ can be formulated as Eq.(7):
>
> $\xi \leq \sqrt{\sum_k |\lambda_k(\Theta_{\theta_{0}})|^2} + \sqrt{\sum_k |\lambda_k(\hat{\Theta_{\theta_{0}}})|^2}.$ (7)
>
> Based on Eq.(4) and Eq.(7), we can conclude that there exists a margin $\xi$ satisfying $0 \leq \xi \leq \sqrt{\sum_k |\lambda_k(\Theta_{\theta_{0}})|^2} + \sqrt{\sum_k |\lambda_k(\hat{\Theta_{\theta_{0}}})|^2}$ such that $\lambda_{min}(\hat{\Theta_{\theta_{0}}})$ and $\lambda_{min}(\Theta_{\theta_{0}})$ negatively correlated. Therefore, the validity of approximating $-\lambda_{min}(\hat{\Theta_{\theta_{0}}})$ with $\lambda_{min}(\Theta_{\theta_{0}})$ is justified.
>
> In this revision, the above theoretical justification is added to **Section A.2 of Appendix**.

---

> ### Author Response · Authors · 2024-11-22
> **Response to Reviewer kdEp [Part 2/3]**
>
> > **The performance of the black-box and white-box attacks is demonstrated using a single dataset (CIFAR-10) in Tables 1 and 2. The authors should consider providing performance comparisons across multiple datasets for the chosen baselines in both tables. Additionally, Table 2 should clearly present a comparison of search efficiency.**
>
> We totally agree with your suggestion, and have provided performance comparisons on CIFAR-100 and ImageNet under white-box and black-box attacks, along with the comparison of search efficiency in Table 2.
>
> + In terms of the comparison under white-box attacks, we choose the models in Table 1 as the peer competitors, and the results of the performance comparison are shown in the following two tables. As can be seen, our derived architectures achieve the state-of-the-art adversarial robustness among all the models chosen for comparison. Meanwhile, our derived architectures achieve the second-best and the best natural accuracy on CIFAR-100 and ImageNet, respectively. These results demonstrate that the proposed zero-cost proxy is effective across multiple datasets under white-box attacks. In this revision, these results and the corresponding summarizations are added to **Section B.1 of Appendix**.
>
> **Performance comparisons on CIFAR-100 under white-box attacks:**
>
> |Models|Natural Acc.|FGSM|PGD$^{20}$|PGD$^{100}$|AA|
> |-|-|-|-|-|-|
> |ResNet-18|55.12%|25.65%|21.08%|19.98%|18.02%|
> |DenseNet-121|**61.71%**|34.28%|27.30%|27.07%|24.55%|
> |DARTS|59.14%|30.35%|25.66%|25.40%|22.65%|
> |PDARTS|58.41%|34.81%|29.11%|28.87%|24.07%|
> |RACL|59.18%|32.04%|26.61%|26.20%|22.92%|
> |DSRNA|57.44%|35.03%|28.11%|27.97%|25.20%|
> |WsrNet|57.81%|28.08%|23.27%|23.01%|21.57%|
> |GradNorm|58.66%|32.87%|28.33%|28.05%|25.58%|
> |SynFlow|55.66%|30.08%|25.24%|24.90%|22.46%|
> |CroZe|59.23%|30.31%|26.16%|26.02%|22.82%|
> |**Ours**|59.39%|**35.73%**|**32.10%**|**31.88%**|**29.95%**|
>
> **Performance comparisons on ImageNet under white-box attacks:**
>
> |Models|Natural Acc.|FGSM|PGD$^{20}$|PGD$^{100}$|AA|
> |-|-|-|-|-|-|
> |ResNet-18|47.38%|17.88%|8.88%|8.37%|7.91%|
> |DenseNet-121|44.13%|12.51%|3.74%|3.14%|3.72%|
> |DARTS|50.58%|17.45%|10.07%|9.44%|8.53%|
> |PDARTS|51.56%|18.10%|10.18%|9.40%|8.77%|
> |RACL|51.59%|18.15%|10.49%|9.82%|8.99%|
> |DSRNA|43.32%|13.04%|7.88%|7.49%|6.47%|
> |WsrNet|50.93%|17.58%|10.15%|9.48%|8.77%|
> |GradNorm|51.34%|18.30%|10.71%|9.90%|9.25%|
> |SynFlow|51.47%|18.90%|10.73%|9.93%|9.42%|
> |CroZe|49.95%|16.54%|9.67%|9.10%|8.36%|
> |**Ours**|**52.71%**|**19.88%**|**11.96%**|**10.94%**|**10.38%**|
>
> + In terms of the comparison under black-box attacks, we choose the models in Table 2 and two more recent models (i.e., WsrNet and CroZe) for the performance comparison on CIFAR-100 and ImageNet. The experimental results are shown in the following two tables. As can be seen, our model achieves the highest attack success rate (100% - the test accuracy, highlighted in bold) for all the target models except RACL on CIFAR-100. Meanwhile, our model also achieves the highest attack success rate for all the models on ImageNet. Furthermore, when our model is set as the target model, it demonstrates the best adversarial robustness among all the baselines chosen. These results demonstrate that the proposed zero-cost proxy is effective across multiple datasets under black-box attacks. In this revision, these results and the corresponding summarizations are added to **Section B.3 of Appendix**.
>
> **Performance comparisons on CIFAR-100 under black-box attacks:**
>
> |Source\Target|DSRNA|RACL|AdvRush|WsrNet|CroZe|Ours|
> |-|-|-|-|-|-|-|
> |DSRNA|-|41.74%|45.04%|42.73%|43.13%|45.72%|
> |RACL|42.90%|-|44.04%|42.84%|44.33%|46.87%|
> |AdvRush|41.83%|**39.60%**|-|41.66%|43.30%|45.53%|
> |WsrNet|44.66%|43.30%|47.13%|-|46.38%|47.68%|
> |CroZe|41.98%|41.68%|45.06%|43.34%|-|46.15%|
> |Ours|**39.90%**|40.43%|**43.08%**|**40.89%**|**41.76%**|-|
>
> **Performance comparisons on ImageNet under black-box attacks:**
>
> |Source\Target|DSRNA|RACL|AdvRush|WsrNet|CroZe|Ours|
> |-|-|-|-|-|-|-|
> |DSRNA|-|31.50%|31.74%|29.38%|27.97%|36.69%|
> |RACL|23.77%|-|27.48%|27.49%|26.51%|32.72%|
> |AdvRush|23.65%|27.22%|-|27.37%|26.23%|32.54%|
> |WsrNet|22.45%|27.97%|28.13%|-|25.75%|33.40%|
> |CroZe|22.41%|28.14%|28.17%|26.94%|-|33.71%|
> |Ours|**21.72%**|**24.43%**|**24.49%**|**24.61%**|**23.82%**|-|
>
> + In this revision, the search efficiency has been compared and added to Table 2. As shown in the table below, the search cost of the proposed zero-cost proxy is significantly lower than that of the peer competitors in Table 2.
>
> |Models|Search Cost (GPU days)|
> |-|-|
> |DSRNA|0.4|
> |RACL|0.5|
> |AdvRush|0.7|
> |**Ours**|**0.017**|

---

> ### Author Response · Authors · 2024-11-22
> **Response to Reviewer kdEp [Part 3/3]**
>
> > **I am curious about the rationale for selecting only specific models (ResNet-18 and PDARTS) for comparison in Table 3. Would it be possible to include models from Table 1 to evaluate their transferability in Table 3?**
>
> Thank you for pointing this out. The reason for choosing specific models (i.e., ResNet-18 and PDARTS) mainly comes from two aspects. First, it is a convention in the robust NAS community to only select specific models for the evaluation of transferability instead of all the peer competitors [2], [3]. Second, ResNet-18 and PDARTS are commonly selected models for the comparison in terms of the transferability [2], [3].
>
> In response to your comment, we have taken models from Table 1 to evaluate their transferability on SVHN and Tiny-ImageNet-200. Specifically, these models are directly transferred and adversarially trained on both datasets for evaluation. As can be seen from the following two tables, comparing with these models, our model still achieves decent natural accuracy and adversarial robustness, demonstrating superior transferability. In this revision, these results and the corresponding summarizations are added to **Section B.4 of Appendix**.
>
> **Experimental results of transferability on SVHN:**
>
> |Model|Natural Acc.|FGSM|PGD$^{20}$|
> |-|-|-|-|
> |DenseNet-121|93.72%|89.68%|72.62%|
> |DARTS|94.90%|90.01%|77.58%|
> |RobNet-free|92.45%|89.33%|85.30%|
> |DSRNA|91.58%|91.27%|84.94%|
> |WsrNet|94.97%|76.67%|84.20%|
> |GradNorm|95.16%|92.51%|90.53%|
> |SynFlow|95.52%|91.53%|76.96%|
> |CroZe|93.19%|66.36%|48.11%|
> |**Ours**|**95.79%**|**95.14%**|**91.64%**|
>
> **Experimental results of transferability on Tiny-ImageNet-200:**
>
> |Model|Natural Acc.|FGSM|PGD$^{20}$|
> |-|-|-|-|
> |DenseNet-121|46.26%|22.88%|19.11%|
> |DARTS|45.94%|24.36%|21.74%|
> |RobNet-free|44.24%|25.44%|23.85%|
> |DSRNA|44.42%|28.52%|**24.32%**|
> |WsrNet|48.62%|22.65%|19.86%|
> |GradNorm|49.17%|16.02%|11.35%|
> |SynFlow|50.96%|12.80%|8.13%|
> |CroZe|48.06%|10.17%|6.08%|
> |**Ours**|**53.92%**|**32.81%**|15.72%|
>
> >**There is inadequate justification for why their approach exhibits better transferability in Table 3. It would be beneficial to provide justification grounded in their theoretical contributions, particularly highlighting the unique advantages that their proposed techniques offer to enhance transferability.**
>
> Thanks for the valuable suggestion. The transferability of the proposed zero-cost proxy can be attribute to the design of the zero-cost proxy. In particular, when evaluating the adversarial robustness $R$ of the architecture, the input samples $x$ are not specified. Meanwhile, the initial weights $\theta_0$ of the network is randomly determined, without any specific constraints. Consequently, the proposed zero-cost proxy can find the inherently robust neural architectures, without the dependence on specific data or weights. As a result, the architecture derived on a certain dataset based on the proposed zero-cost proxy can be well transferred to other datasets. In this revision, the above justification is added to **Section B.4 of Appendix**.
>
> >**The authors should discuss the assumptions made in their analysis and their validity within the context of the study. For instance, when transforming from Equation 1 to Equation 3, it would be helpful to clarify the assumptions involved and how they apply in their setting. Specifically, I am uncertain whether the assumption of infinite-width DNN parameters is valid; if it is not, what implications does that have for this paper? I encourage the authors to include these details, even if they are standard, as doing so would enhance the paper's readability and facilitate the assessment process.**
>
> Thank you for the nice suggestion. We have discussed the assumption in terms of the infinite-width DNN parameters. In particular, when transferring Eq.(1) to Eq.(3), the assumption of infinite-width DNN parameters is valid. As evidenced by the previous literature [4], in the infinite-width limit, the NTK becomes deterministic at initialization and stays constant during training. Consequently, when replacing $\Theta_{\theta_t}$ in Eq.(1) with $\Theta_{\theta_0}$ in Eq.(3), because of the invariance of NTK in the infinite-width limit, these two NTKs can be directly replaced by each other. Therefore, the transformation from Eq.(1) to Eq.(3) keeps valid for the infinite-width DNN parameters. In this revision, the above discussions are added to **Section A.1 of Appendix**.
>
> **References**
>
> [1] Demystifying the neural tangent kernel from a practical perspective: Can it be trusted for neural architecture search without training?, CVPR 2022.
>
> [2] When NAS meets robustness: In search of robust architectures against adversarial attacks, CVPR 2020.
>
> [3] AdvRush: Searching for Adversarially robust neural architectures, ICCV 2021.
>
> [4] Neural tangent kernel: Convergence and generalization in neural networks, NeurIPS 2018.

---

> > ### Comment · Reviewer_kdEp · 2024-11-27
> > **Thank you for the detailed rebuttal: Raised the Score**
> >
> > Dear Authors,
> > I would like to express my sincere gratitude for the thorough and meticulously prepared rebuttal you have provided. I am satisfied with the clarifications and explanations offered in the rebuttal. As a result, I have increased the score to 6. Unfortunately, we do not have the option to raise the score to 7 but if an option were available, I would do so. Thanks.

---

> > > ### Author Response · Authors · 2024-11-27
> > > **Response to Reviewer kdEp**
> > >
> > > Thank you for your response and for upgrading your score. We sincerely appreciate the time and effort you dedicated to reviewing our work. Your constructive comments and suggestions are valuable in helping us improve the paper.

---

### Official Review · Reviewer_jcgN · 2024-11-04

**Soundness:** 3
**Presentation:** 3
**Contribution:** 4
**Rating:** 6
**Confidence:** 3

**Summary:**

This paper proposes a zero-shot proxy that requires no training and depends only on the initial neural network weights to find the robust architecture. Using this zero-shot proxy reduces the neural architecture search speed drastically. Experiments are performed on multiple datasets with varying resolution and number of classes.

**Strengths:**

- The reduction in the search cost is very significant.
- The paper is well-written with typos.
- From Table 3, the transferability results are good.
- A novel dataset will be released by the authors, which will help advance research in the robust NAS direction.
- Comparison with the latest state-of-the-art methods.

**Weaknesses:**

- While the search cost is reduced significantly, the increase in the adversarial robustness is minimal (less than 1% in most cases for Table 1). A similar observation can be derived from the results in Table 4.
- Minor typos: "8/2550" instead of "8/255" on line 301, "We" instead of "we" on line 63.

**Questions:**

While the contribution in terms of speed reduction is significant for the proposed method, my concern is the increase in adversarial robustness as compared to the baselines is very minimal in the majority of the cases.

---

> ### Author Response · Authors · 2024-11-22
> **Response to Reviewer jcgN**
>
> Thank you sincerely for the encouragement and the recognition in our work. We appreciate your great effort and address your comments in detail below.
>
> > **While the contribution in terms of speed reduction is significant for the proposed method, my concern is the increase in adversarial robustness as compared to the baselines is very minimal in the majority of the cases (less than 1% in most cases for Tables 1 and 4).**
>
> Thank you for your insightful comment. We address your concern from two aspects:
>
> + First, please kindly note that the main contribution of this paper is to **accelerate the adversarial robustness evaluation** while maintaining the adversarial robustness. As can be seen from Table 1, the proposed zero-cost proxy achieves more than 20$\times$ speedup comparing with the state-of-the-art robust NAS methods, and the adversarial robustness also demonstrates slight improvement. Based on these results, the main contribution of this work has already been well supported even though the increase in adversarial robustness is not significant.
>
> + Second, we have provided additional experimental results on CIFAR-100 beyond CIFAR-10 (Table 1) and ImageNet (Table 4). As shown in the table below, the proposed zero-cost proxy achieves the improvements of 3%-4% under the PGD$^{20}$, PGD$^{100}$, and AA attacks, demonstrating the proposed zero-cost proxy is still potential for a relatively large increase in adversarial robustness. The results in the table below and the corresponding summarizations have been added to **Section B.1 of Appendix** in this revision.
>
> |Models|FGSM|PGD$^{20}$|PGD$^{100}$|AA|
> |-|-|-|-|-|
> |ResNet-18|25.65%|21.08%|19.98%|18.02%|
> |DenseNet-121|34.28%|27.30%|27.07%|24.55%|
> |DARTS|30.35%|25.66%|25.40%|22.65%|
> |PDARTS|34.81%|29.11%|28.87%|24.07%|
> |RACL|32.04%|26.61%|26.20%|22.92%|
> |DSRNA|35.03%|28.11%|27.97%|25.20%|
> |WsrNet|28.08%|23.27%|23.01%|21.57%|
> |GradNorm|32.87%|28.33%|28.05%|25.58%|
> |SynFlow|30.08%|25.24%|24.90%|22.46%|
> |CroZe|30.31%|26.16%|26.02%|22.82%|
> |**Ours**|**35.73%**|**32.10%**|**31.88%**|**29.95%**|
>
> > **Minor typos: "8/2550" instead of "8/255" on line 301, "We" instead of "we" on line 63.**
>
> Thank you for your meticulous review. In this revision, we have fixed the mentioned typo in line 63. As for the word "8/255" in line 301, please kindly note that this is the correct setting of our experiments instead of a typo. Specifically, the total perturbation scale of the PGD$^7$ attack is set to 8/255 by following the convention [1]. Meanwhile, the step size of each step in the PGD attack is set to 8/2550, which also follows the convention [1] to ensure the fair comparison.
>
> **Reference**
>
> [1] Generalizable lightweight proxy for robust NAS against diverse perturbations, NeurIPS 2023.

---

> > ### Comment · Reviewer_jcgN · 2024-11-26
> > **Response to rebuttal**
> >
> > Thank you for addressing my concerns. I would suggest the authors highlight their focus (like they did in the rebuttal) on accelerating adversarial robustness in the paper, too.

---

> > > ### Author Response · Authors · 2024-11-27
> > > **Response to Reviewer jcgN**
> > >
> > > Thank you for your response and valuable suggestion. We have highlighted our focus on accelerating adversarial robustness evaluation in lines 50-52 in the revised version.

---

### Official Review · Reviewer_rBjP · 2024-11-04

**Soundness:** 3
**Presentation:** 2
**Contribution:** 3
**Rating:** 6
**Confidence:** 4

**Summary:**

This paper targets a lightweight proxy to assess the adversarial robustness of Neural Architecture Search (NAS) networks. The proposed proxy is represented as the product of two terms (Eq. 4). The first term, based on the intuition that adversarial and natural accuracy are reversely correlated, is introduced to consider adversarial accuracy and is experimentally validated. The second term is proposed through a theoretical approximation using the Neural Tangent Kernel. The proposed proxy is experimentally validated.

**Strengths:**

- The paper is easy to read through, well presented.
- The paper proposes relatively strong method for the proxy of searching adversarially robust architectures.
- The paper includes numerous validations including performance under white-box, black-box attacks, datasets, numerous datasets, and ablation studies.

**Weaknesses:**

1.  To read through, I felt that the paper has to focus on two points :
- Is the proposed method superior in terms of cost-efficiency? (zero-cost)
- Is the proposed method superior in terms of robust accuracy?
- For now, RQ1 and RQ2 looks a bit similar(line 255-256), before looking at the experimental results.

2. I feel the cost-efficiency related explanation is slightly short. The authors could include detailed explanations for (line 183-185.) such as , the reason the authors chose to iterate samples over generating adversarial samples itself and why is it efficient.

**Questions:**

- In the preliminatries, 3.1., the formulation is bit awkward; if the function $f_{\theta_t}$ is a function $\mathbb{R}^d \rightarrow \mathbb{R}$, then the output will be a single scalar, but in the equations the l2-norm calculation is applied … I also checked the reference paper (Xu et al., (2021)), but it is slightly different.

- The overall experiments on Kendall’s tau seems to be not high (Fig.2,Fig.4). (Relatively high, but not significant in absolute terms). Can the authors explain this in further details?

- The cost analysis is done with the GPU search day metric. How is the cost calculated? Under which GPU?

[Post-Rebuttal] I have reviewed the author's rebuttal and the feedback from other reviewers. Most of my concerns have been addressed, and given the paper's contributions, I would like to maintain my current rating.

---

> ### Author Response · Authors · 2024-11-22
> **Response to Reviewer rBjP**
>
> We sincerely thank you for the time and effort you have invested in reviewing our paper. Your comments and questions are very insightful and highly valuable for us to enhance the paper. Your concerns are addressed point-by-point in the following.
>
> ## Weaknesses
>
> > **RQ1 and RQ2 looks a bit similar before looking at the experimental results.**
>
> Actually, the two points (i.e., the superiority in terms of cost-efficiency and robust accuracy) mentioned by the reviewer are both included in experiments for RQ1. Specifically, RQ1 mainly focus on the effectiveness of the proposed zero-cost proxy in terms of **searching for architectures**. To demonstrate the superiority of the proposed zero-cost proxy in this setting, both cost-efficiency and robust accuracy of the derived architecture need to be contained following the convention [1]. Moreover, RQ2 mainly focus on the superiority of the proposed zero-cost proxy in terms of **evaluating the adversarial robustness of architectures**, which is a different emphasis than that of RQ1.
>
> To avoid the confusion, we have revised RQ1 and RQ2 in **lines 260 and 261** for further clarification:
>
> **RQ1**: Is the proposed zero-cost proxy superior in robust accuracy and cost-efficiency?
>
> **RQ2**: Is the proposed zero-cost proxy superior in evaluating adversarial robustness of architectures?
>
> > **The cost-efficiency related explanation is slightly short.**
>
> In response to this comment made by the reviewer, we have explained the reason for choosing to iterate samples over generating adversarial examples. Specifically, the NTK is calculated only based on different samples $x$ and $x'$ as shown in Equation (2). This means that we only need to ensure the samples are different, and it does not matter whether they are natural data or adversarial examples. Consequently, we can obtain the different samples by simply iterating samples in a batch, without the need to generate adversarial examples. Because simply iterating samples does not introduce much computational cost, and the repetitive calculation for the gradient or the time-consuming optimization for the adversarial example generation are no longer needed, the proposed zero-cost proxy demonstrates superior efficiency. These detailed explanations have been added to **Section 3.2 (lines 182-190)** in this revision.
>
> ## Questions
>
> > **The formulation is a bit awkward in Section 3.1.**
>
> We are sorry for confusing the reviewer on this point. Actually, the output of the function $f_{\theta_t}$ is a **prediction label vector** instead of a single scalar. This is because the input $x$ of $f_{\theta_t}$ is a batch of image samples. Suppose the number of samples is $N$, $x$ can be denoted as $\{x_1, x_2, \ldots, x_N\}$, and the output of $f_{\theta_t}$ can be denoted as $\{f_{\theta_t}(x_1), f_{\theta_t}(x_2), \ldots, f_{\theta_t}(x_N)\}$, which is a prediction label vector. Then, the $L_2$ distance between the output of $f_{\theta_t}$ and the corresponding label vector $\{y_1, y_2, \ldots, y_N\}$ of $x$ is calculated.
>
> In this revision, we have added the clarification "$x$ and $y$ denote image samples and the corresponding label vector" in **line 145** to avoid the confusion.
>
> > **The Kendall's tau seems to be not high in Figures 2 and 4.**
>
> The phenomenon mentioned by the reviewer can be explained by the different size of the architectural benchmarks. As can be seen from Figure 2, the highest value of Kendall's tau reaches 0.47 under the FGSM attack on the NAS-Bench-201-R benchmark (containing 6,466 architectures naturally trained). Furthermore, because the Tiny-RobustBench benchmark (containing 223 architectures adversarially trained) is smaller than NAS-Bench-201-R, it cannot represent the distribution of all architectures in the search space very exactly. As a result, the value of the Kendall's tau becomes lower. However, the Kendall's tau of the proposed zero-cost proxy is still the highest among all the competitors.
>
> In addition, the above claim can also be supported by the recent study [2]. The experimental results in this study show that when using the whole NAS-Bench-201 (containing 15,625 architectures naturally trained) to test the zero-cost proxies in terms of evaluating the natural accuracy, the value of Kendall's tau becomes larger than 0.5. Consequently, it is further demonstrated that larger benchmarks lead to higher values of the Kendall's tau in absolute terms.
>
> > **How is the GPU day metric calculated? Under which GPU?**
>
> We are sorry for not containing these details in the initial version. The GPU days metric is calculated by "number of GPUs used $\times$ total running time (days)". The type of GPU used for the cost analysis is NVIDIA RTX 2080Ti. These details are added to lines 283-284 in this revision.
>
> **References**
>
> [1] Generalizable lightweight proxy for robust NAS against diverse perturbations, NeurIPS 2023.
>
> [2] DisWOT: Student architecture search for distillation without training, CVPR 2023.

---

> > ### Comment · Reviewer_rBjP · 2024-11-27
> > **Response to rebuttal**
> >
> > Dear authors, I have reviewed the author's rebuttal and the feedback from other reviewers. Most of my concerns have been addressed, and given the paper's contributions, I would like to maintain my current rating.

---

> > > ### Author Response · Authors · 2024-11-27
> > > **Response to Reviewer rBjP**
> > >
> > > We sincerely appreciate the valuable time and effort you invested in reviewing our work. Your comments have been very helpful in improving the paper.

---

### Author Response · Authors · 2024-11-22
**General Response**

We wholeheartedly thank all reviewers for the great effort and the meticulous review. The comments and suggestions are truly impressive and helpful to enhance our paper. In response to the comments and suggestions, we have made the following updates in this revision:

+ The typo in line 63 has been fixed. [jcgN]

+ The description **in the preliminaries 3.1** is updated, in order to avoid the confusion in terms of the $L_2$-norm calculation. [rBjP]

+ More detailed explanations and discussions in terms of the cost-efficiency are added to **lines 182-190 in Section 3.1**, in order to further explain why the proposed zero-cost proxy enjoys decent efficiency. [rBjP]

+ The contents of RQ1 and RQ2 are revised to further distinguish the core points of both research questions. [rBjP]

+ The details in terms of the calculation of the GPU days metric are added to **lines 283-284 in Section 4.1.1**. [rBjP]

+ The discussion about the assumption applied to the proposed zero-cost proxy is added to **Section A.1 of Appendix**, in order to enhance the paper's readability. [kdEp]

+ The theoretical justification in terms of the validity of approximating $-\lambda_{min} (\hat{\Theta_{\theta_{0}}})$ with $\lambda_{min}(\Theta_{\theta_{0}})$ is added to **Section A.2 of Appendix** to enhance the theoretical robustness of the proposed zero-cost proxy. [kdEp]

+ Additional experimental results across various datasets under both white-box and black-box attacks are presented in **Sections B.1 and B.3 of Appendix**, respectively, to make the comparisons more comprehensive. [kdEp]

+ The results of more baseline models in terms of the transferability are added to **Section B.4 of Appendix**, along with the justification for the transferability grounded in the theoretical contributions of this paper. [kdEp]

Please note that the revised contents are highlighted in red in this version for the reviewers' convenience.

---

### Meta-Review · Area_Chair_kcGZ · 2024-12-19

**Metareview:**

All reviewers agree that the paper is (marginally) above the acceptance threshold. They highlight the proposed zero-cost proxy as a novel approach for searching adversarially robust architectures, with experimental results effectively demonstrating its efficacy. Additionally, the introduced dataset is considered a valuable contribution to the research community.
Initially, reviewers raised concerns regarding certain experimental settings, results, and methodological details. After the authors' rebuttal, all reviewers acknowledged that most concerns were adequately addressed. The recommendation is to accept the paper.

**Additional Comments On Reviewer Discussion:**

Initially, reviewers raised concerns regarding certain experimental settings, results, and methodological details. After the authors' rebuttal, all reviewers acknowledged that most concerns were adequately addressed.

---

### Decision · Program_Chairs · 2025-01-22

Accept (Poster)